# A slowly cleaved viral signal peptide acts as a protein-integral immune evasion domain

Einat Seidel [1], Liat Dassa[1], Shira Kahlon[1], Boaz Tirosh [2], Anne Halenius[3,4], Tal Seidel Malkinson[5] & Ofer Mandelboim [1]✉

Stress can induce cell surface expression of MHC-like ligands, including MICA, that activate NK cells. Human cytomegalovirus (HCMV) glycoprotein US9 downregulates the activating immune ligand MICA*008 to avoid NK cell activation, but the underlying mechanism remains unclear. Here, we show that the N-terminal signal peptide is the major US9 functional domain targeting MICA*008 to proteasomal degradation. The US9 signal peptide is cleaved with unusually slow kinetics and this transiently retained signal peptide arrests MICA*008 maturation in the endoplasmic reticulum (ER), and indirectly induces its degradation via the ER quality control system and the SEL1L-HRD1 complex. We further identify an accessory, signal peptide-independent US9 mechanism that directly binds MICA*008 and SEL1L. Collectively, we describe a dual-targeting immunoevasin, demonstrating that signal peptides can function as protein-integral effector domains.

[1] The Lautenberg Center for General and Tumor Immunology, The Faculty of Medicine, The Hebrew University Medical School, IMRIC, Jerusalem, Israel. [2] The Institute for Drug Research, Hebrew University Faculty of Medicine, Hebrew University of Jerusalem, Jerusalem, Israel. [3] Institute of Virology, Medical Center University of Freiburg, Freiburg, Germany. [4] Faculty of Medicine, University of Freiburg, Freiburg, Germany. [5] Paris Brain Institute, Sorbonne Université, Inserm UMRS 1127, CNRS UMR 7225, Paris, France. ✉email: oferm@ekmd.huji.ac.il

N-terminal cleavable signal peptides (SP) direct nascent soluble and type-I transmembrane (TM) polypeptides into the endoplasmic reticulum (ER) and the secretory pathway[1–3]. SP sequences vary in composition and length, but all have polar regions flanking a hydrophobic core which inserts into the ER membrane[3–5]. The cleavage site is determined by small uncharged residues in positions −3 and −1[6]. Cleavage is mediated by signal peptide peptidase and generally occurs co-translationally, though a few exceptions were reported[3–5].

The ER provides an environment suitable for proper protein folding and maturation. In it, proteins undergo N-linked glycosylation, disulfide bond formation, and attachment of glycosyl-phosphatidylinositol (GPI) membrane anchors[7,8]. When a protein in the ER fails to fold correctly, it is recognized by the ER quality control (ERQC) machinery. N-linked glycosylations play an essential role in glycoprotein quality control: glycan composition signals whether a protein is misfolded and should be retained in the ER for additional folding attempts. Furthermore, glycans serve as timers which limit such folding attempts. Specifically, ER mannosidases progressively trim N-linked glycosylations until they become too short to support further folding attempts[9]. Lectins then deliver the terminally misfolded protein to an E3-ligase complex that retrotranslocates the misfolded protein from the ER to the cytosol for proteasomal degradation, a process called ER-associated degradation (ERAD)[10].

Human cytomegalovirus (HCMV) is a herpesvirus which persistently infects most of the human population. It is the leading infectious cause of birth defects in the developed world, and poses significant risks to immunosuppressed patients[11]. Its complex dsDNA genome encodes hundreds of proteins and several classes of non-coding RNAs[12,13]. A significant portion of its genome is devoted to evading immune responses, ensuring viral persistence for the host's lifetime[14]. A hallmark of HCMV immune evasion is manipulation of the ERAD and secretory pathways to prevent cell surface expression of activating immune ligands and subsequent recognition by immune cells[15].

Natural killer (NK) cells constitute a major HCMV immune evasion target[14]. Best known for their ability to kill cancer cells and virally infected cells[16], NK cells are crucial in controlling HCMV infections[17]. NK cells are also important sources of cytokines and chemokines[16]. NK cells express a wide array of activating and inhibitory receptors, and are triggered when signal summation tips toward activation[18]. To counter NK cell activation, HCMV upregulates the expression of cellular and viral inhibitory NK ligands, while simultaneously suppressing the expression of activating NK ligands[14,19].

A key activating ligand targeted by HCMV is major histocompatibility complex (MHC) class I polypeptide-related sequence A (MICA). MICA is part of a family of eight stress-induced MHC-like ligands, along with MICB and UL16-binding proteins 1–6 (ULBP1-6)[20]. These ligands are upregulated in response to stress such as DNA damage and viral infection. The activating NK receptor NKG2D binds these ligands and facilitates the elimination of dangerous cells[20]. MICA is the most polymorphic member of the family, with about 100 known alleles[21]. Of these alleles, MICA*008 is one of the most prevalent[21], and has unique biological properties. Due to a frameshift mutation in its transmembrane domain, MICA*008 is first synthesized as a truncated soluble protein. It then remains in the ER until it becomes GPI-anchored, permitting ER egress and surface expression. The slow process that directs MICA*008 to GPI anchoring despite its lack of a canonical GPI-anchoring signal is poorly characterized, and the factors involved remain unknown[22]. In contrast, non-truncated MICA alleles are transmembrane proteins and follow a different and faster maturation pathway[22].

Several HCMV immune evasion proteins inhibit MICA expression. US18 and US20 act together to send MICA to lysosomal degradation[23], as does UL148A, which acts in tandem with an unknown viral binding partner[24]. Finally, UL142 retains MICA in the *cis*-Golgi apparatus[25–27]. The truncated allele MICA*008 is, however, resistant to all of these mechanisms[24,25]. Consequently, until recently MICA*008 was considered an example of human adaptation to viral selective pressure[14]. In our previous work, we found that MICA*008 is in fact susceptible to HCMV[28]. A viral glycoprotein called US9 selectively targets MICA*008 for proteasomal degradation, hampering NKG2D-mediated NK killing of HCMV-infected cells. Interestingly, US9 function seems to depend on MICA*008's non-standard maturation pathway. US9 degrades MICA*008 with slow kinetics, after a prolonged lag in the ER but prior to the GPI-anchoring step. Moreover, US9 function is only possible when MICA*008 undergoes non-canonical GPI-anchoring, since substituting the frameshifted MICA*008 TM domain with a canonical GPI-anchoring signal confers resistance to US9[28].

Here, we explore US9 structure-function relationship and underlying mechanisms of action. We show that the US9 SP is cleaved with unusually slow kinetics. We establish that the US9 SP is the major US9 domain targeting MICA*008 to degradation, thanks to this slow cleavage. We further show that US9 contains a second, SP-independent mechanism targeting MICA*008 that directly interacts with MICA*008 and the SEL1L-HRD1 ERAD complex. Last, we show that the US9 SP arrests MICA*008 maturation and indirectly induces ERQC-mediated degradation, also via the SEL1L-HRD1 ERAD complex.

## Results

**US9 SP is cleaved with unusually slow kinetics.** US9 is a type-I transmembrane ER-resident glycoprotein of approximately 30 kDa, expressed early in HCMV infection[13,29–31]. In gel electrophoresis, US9 migrates as two distinct protein bands, previously ascribed to differential glycosylation[31]. Hence, we expected that endoglycosidase H (endoH), an enzyme that digests the high mannose N-linked glycans of ER-resident glycoproteins, would cause US9 to run as a single band. We used previously described[28] RKO cells which endogenously express very low levels of the full-length allele MICA*007:01, and overexpress MICA*008 fused to an N-terminal HA tag. We transduced the cells with an empty vector (EV) or US9. However, when we performed endoH digestion followed by immunoblotting, we observed that deglycosylated US9 still migrated as a doublet (Fig. 1a), suggesting that a difference in the polypeptide backbone itself is responsible for these two forms. The difference could not be at the C-terminus since the US9 construct we used bears a C-terminal 6XHis tag and cleavage of that end would have prevented recognition by the anti-tag antibody, so we suspected that N-terminal processing caused the observed size difference. The approximately 3 kDa size difference between the two US9 forms matched the predicted size of the N-terminal SP of 27 amino acids (AA) (see annotated US9 sequence in Fig. S1a). We therefore hypothesized that the two US9 bands represent a SP+ precursor form and a SP− processed form. Of note, a similar two-band appearance caused by delayed N-terminal SP cleavage has been described for US11[32].

To determine the nature of the two US9 forms we conducted a pulse-chase experiment (Fig. 1b and S1b, quantified in Fig. 1c). The larger SP+ form was the first to appear, while the smaller SP− form accumulated very slowly, suggesting a precursor-product relationship. The SP+ form was almost completely gone by 120 min of chase, but its quantity decreased more rapidly than the SP− form accumulated, suggesting rapid turnover of the SP+ form. Accordingly, treatment with the proteasome inhibitor

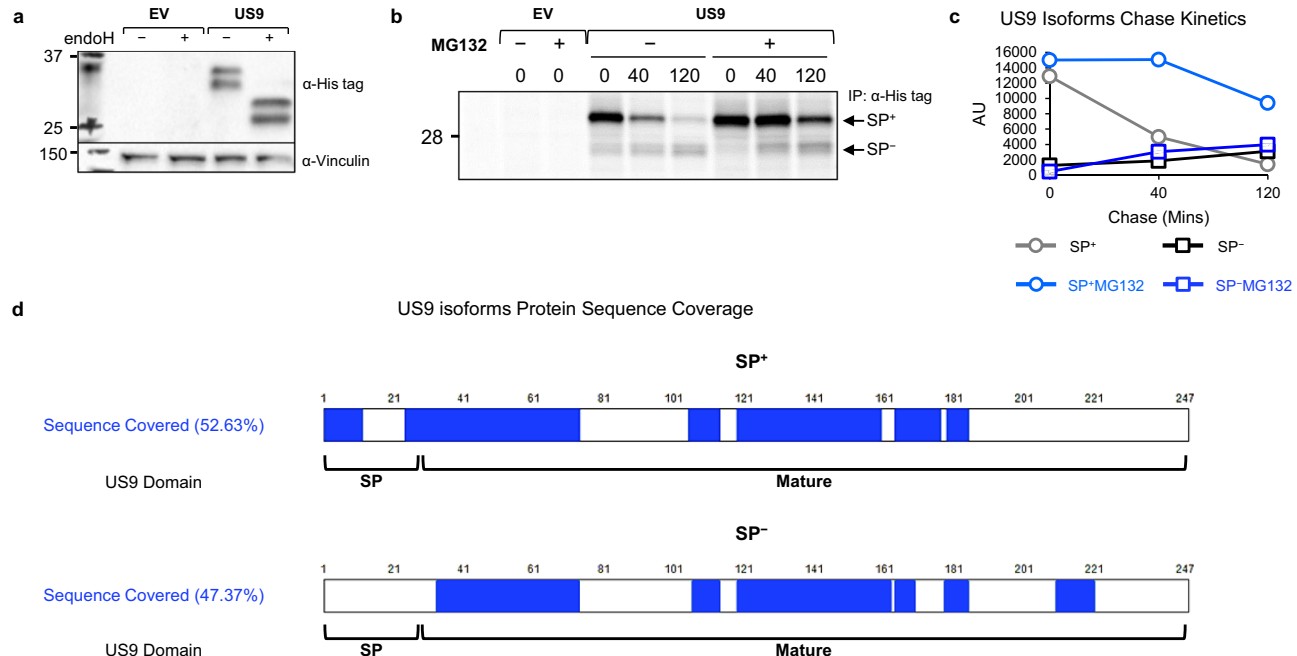

**Fig. 1 US9 contains a slowly cleaved SP. a** RKO MICA*008-HA cells co-expressing an empty vector (EV) or US9 attached to a C-terminal 6XHis tag were lysed. Lysates were mock-treated (−) or digested with endoglycosidase H (endoH; +) and then blotted. Anti-His tag was used to visualize protein expression, with anti-vinculin as loading control. Representative of at least three independent experiments. **b** RKO MICA*008-HA cells expressing EV or US9 were radioactively labeled for 20 min (pulse) and then chased in the absence (−) or presence (+) of 10 μM MG132. Digitonin-lysates were prepared at the indicated chase times (in minutes). An anti-His tag immunoprecipitation was performed. All samples were digested with endoH. Representative of three independent experiments. Ladder was visualized at a higher contrast, shown in Fig. S1b. **c** Quantification of the US9 isoforms shown in (**b**). AU arbitrary units. **d** Graphic representation of US9 protein sequence coverage detected by mass spectrometry analysis of the two US9 isoforms: SP+ and SP−. Sequence coverage % and US9 domains are indicated. See also Supplementary Data 1–2, figure S1 and the Source Data file for experimental data.

MG132 stabilized the SP+ form but did not significantly impact the accumulation of the more stable SP− form. We further validated these results with a cycloheximide (CHX) chase assay (Fig. S1c). Under treatment with the protein translation inhibitor CHX, the SP+ form was initially present, but slowly vanished until it was no longer detectable at 4 h of chase, even in the presence of the proteasome inhibitor epoxomicin (EPX).

To directly assess SP presence, we immunoprecipitated US9, excised each of the two US9 isoforms from an electrophoresis gel, and detected peptides derived from each band using mass spectrometry (Fig. 1d). A sequence coverage rate of about 50% was obtained for both forms, and notably, only the SP+ forms contained peptides derived from the SP, including peptides spanning the predicted cleavage area (Supplementary Data 1–2).

We next addressed the question of the SP cleavage site. Specifically, there are two such predicted sites in the literature: S24[33], and S27[34] (Fig. S1d, residues highlighted in red). We mutated each of these residues to arginine to prevent SP peptidase cleavage[35], creating mutants named S24R and S27R. We then overexpressed the mutants in RKO MICA*008-HA cells and immunoblotted cell lysates. Importantly, S24R remained a doublet, while S27R migrated as a single band the same size as the SP+ form (Fig. S1d). Both mutants were endoH-sensitive (Fig. S1e), ruling out defective ER insertion. These findings confirm S27 is the US9 SP cleavage site. Taken together, our findings show that US9 has two isoforms: the precursor SP+ US9, which is slowly cleaved into the SP− US9 product.

**Generation of US9 mutants.** We previously showed that US9 specifically targets MICA*008 for proteasomal degradation[28]. We next wanted to identify the domains involved in US9 function. Based on Uniprot[34] and Phobius[36] predictions, US9 contains

several putative domains (diagram in Fig. 2a, annotated sequence in Fig. S1a): the N-terminal SP (27 AA), an ER-luminal domain (166 AA), a transmembrane domain (30 AA), and a short cytosolic tail (24 AA). The ER-luminal domain of US9 contains a compositionally biased serine-rich area near the N-terminus (59 AA), and an immunoglobulin (Ig)-like fold (97 AA). Two N-linked glycosylation sites are predicted within the Ig-like fold (N97, N158).

US9 resembles its neighboring genes: US2 and US11[37], which target the MHC class I heavy chain (MHC I) for proteasomal degradation to evade cytotoxic T lymphocytes[38,39]. Structurally, US2 and US11's ER-luminal domains bind MHC I[37,40–43], while their transmembrane (TM) or cytosolic domains recruit specific ERAD complexes[41,43–48]. We therefore hypothesized that US9 might function similarly. We aligned US9 and US11 protein sequences (Fig. S2a) that share 35.7% similarity and found that US9 contains a conserved glutamine (Q214, highlighted in red) in its TM domain. The equivalent glutamine in US11 (Q192) is responsible for recruiting the ERAD complex constituent Derlin-1[41,44,45].

We decided to systematically mutate US9 and perform a structure-function analysis (Fig. 2b). We created three cytosolic and TM domains mutants: ΔCyto+TM—removal of the cytosolic and TM domains; Q214A—conserved glutamine mutated to alanine; and 8TM—the TM domain was swapped with that of US8, a related protein which does not affect MICA*008, recently shown to target TLR3 and TLR4 to degradation[28,49]. We created three luminal deletion mutants: ΔC-lum—deletion of the C-terminal part of the luminal domain up to the Ig-like fold; ΔIg—deletion of the Ig-like fold; and ΔSer—deletion of the serine-rich area up to the 13 N-terminal AA. Last, we created two mutants for the SP and adjacent luminal area: ΔN-Ser—deletion of the 13 N-terminal AA (KESLRLSWSSDES) of the serine-rich area, up to the SP cleavage site; and 8SP—the US9 SP was swapped with the

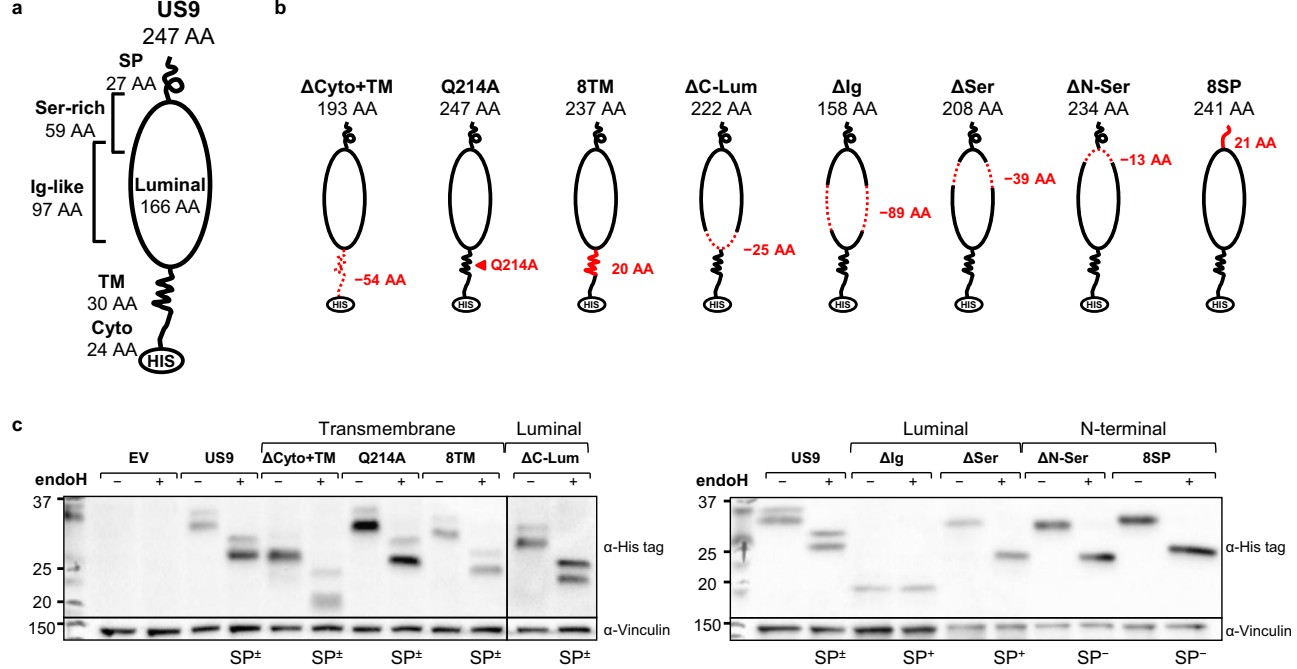

**Fig. 2 Delayed US9 SP cleavage requires the N-Ser area. a** Diagram of US9 domains: signal peptide (SP), serine-rich compositionally biased area (Ser), Ig-like fold (Ig), transmembrane (TM), cytosolic (Cyto), 6XHis tag (HIS). Total protein length without tags and the length of each domain in amino acids (AA) are indicated. **b** Diagrams of US9 deletion and substitution mutants arranged according to the mutated domain. The total length of each construct without tags and of each modified domain are indicated. Dashed red lines show deleted domains, solid red lines show substitutions with a homologous US8 sequence, arrow shows a single mutated residue. **c** The US9 mutants described in (**a**, **b**) were expressed in RKO MICA*008-HA cells. Lysates prepared from the indicated transfectants were untreated (−) or digested with endoH (+) and then immunoblotted. Mutants are grouped by mutated region (indicated). Anti-His tag was used to visualize protein expression, with anti-vinculin as loading control. Annotation beneath the panel indicates deglycosylated US9 forms with, without, or both with and without the SP ($SP^{+/-/\pm}$). Different panels depict separate gels. Different segments of the same gel are shown together for clarity in the left panel. Representative of two independent repeats. See also Fig. S2.

US8 SP to maintain proper ER insertion[33]. All the constructs were fused to a C-terminal 6XHis tag.

**US9 mutants are properly expressed and ER-localized**. The US9 mutants were transduced into RKO MICA*008-HA cells. First, we verified the expression of the US9 mutants by immunoblot (Fig. S2b). All the mutants were expressed, but their abundance varied. We validated ER localization by endoH sensitivity assay (Fig. 2c). Almost all mutants were endoH-sensitive, indicating ER localization. The endoH assay is irrelevant for the ΔIg mutant that lacks N-linked glycosylations. Therefore, we verified by immunofluorescence that like wild type (WT) US9, ΔIg co-localizes with the ER marker PDI (Fig. S2c).

**Delayed cleavage depends on SP sequence and the N-Ser domain**. Intriguingly, the two-band appearance indicating the presence of both $SP^+$ and $SP^-$ forms (named $SP^\pm$ in Fig. 2c) was lost in certain mutants: ΔIg, ΔSer, ΔN-Ser, and 8SP (Fig. 2c, right). This was to be expected for 8SP, since US8 migrates as a single band in immunoblot and pulse-chase experiments, indicating rapid SP cleavage[28,31,49,50]. However, the single band appearance in other mutants suggested that US9 SP cleavage was affected by additional domains. To study the SP status of these mutants, we compared the predicted and observed molecular weights of each deglycosylated mutant. For ΔIg and ΔSer, the observed sizes were compatible with unprocessed forms constitutively retaining their SP (named $SP^+$ in Fig. 2c). In contrast, ΔN-Ser and 8SP sizes were compatible with the predicted processed size (named $SP^-$ in Fig. 2c), indicating rapid cleavage.

We further validated the SP status of the ΔIg, ΔSer, and ΔN-Ser mutants by mass spectrometry, which identified SP-derived sequences in ΔIg and ΔSer, but not in ΔN-Ser (Fig. S2d and Supplementary Data 3–5). Finally, we conducted a pulse-chase experiment comparing US9 and ΔN-Ser maturation kinetics, which showed no evidence of a SP-containing precursor form in the ΔN-Ser mutant (Fig. S2e). This suggested rapid, co-translational SP cleavage occurring before His tag synthesis. Notably, ΔN-Ser was also considerably more stable than $SP^+$ US9.

These results suggest that large luminal domain deletions can prevent US9 SP cleavage, regardless of the deleted domain. In contrast, deletion of the N-Ser area or swapping US9 SP with a rapidly cleaved SP sequence restored rapid SP cleavage. Of note, SP retention inversely correlated with US9 protein abundance: $SP^+$ mutants were less abundant than WT US9, while $SP^-$ mutants were more abundant than WT US9.

**US9 N-terminal mutants are impaired in inducing MICA*008 degradation**. We next assessed US9 mutant functionality. We began by measuring the surface expression of MICA*008 in the presence of each mutant using flow cytometry (Fig. 3a and S2f, quantified in Fig. 3b, summarized in Table 1). Remarkably, all mutants retained some MICA*008 downregulating capacity. Mutating Q214 caused no significant difference compared to WT US9, indicating that this residue is redundant for MICA*008 targeting. The ΔCyto+TM and ΔIg mutations had a statistically significant but relatively small effect on US9 functional capacity; both mutants continued to cause a greater than tenfold reduction in MICA*008 surface levels. Unexpectedly, the two mutants which showed the greatest functional impairment were the N-terminal mutants with rapid SP cleavage,

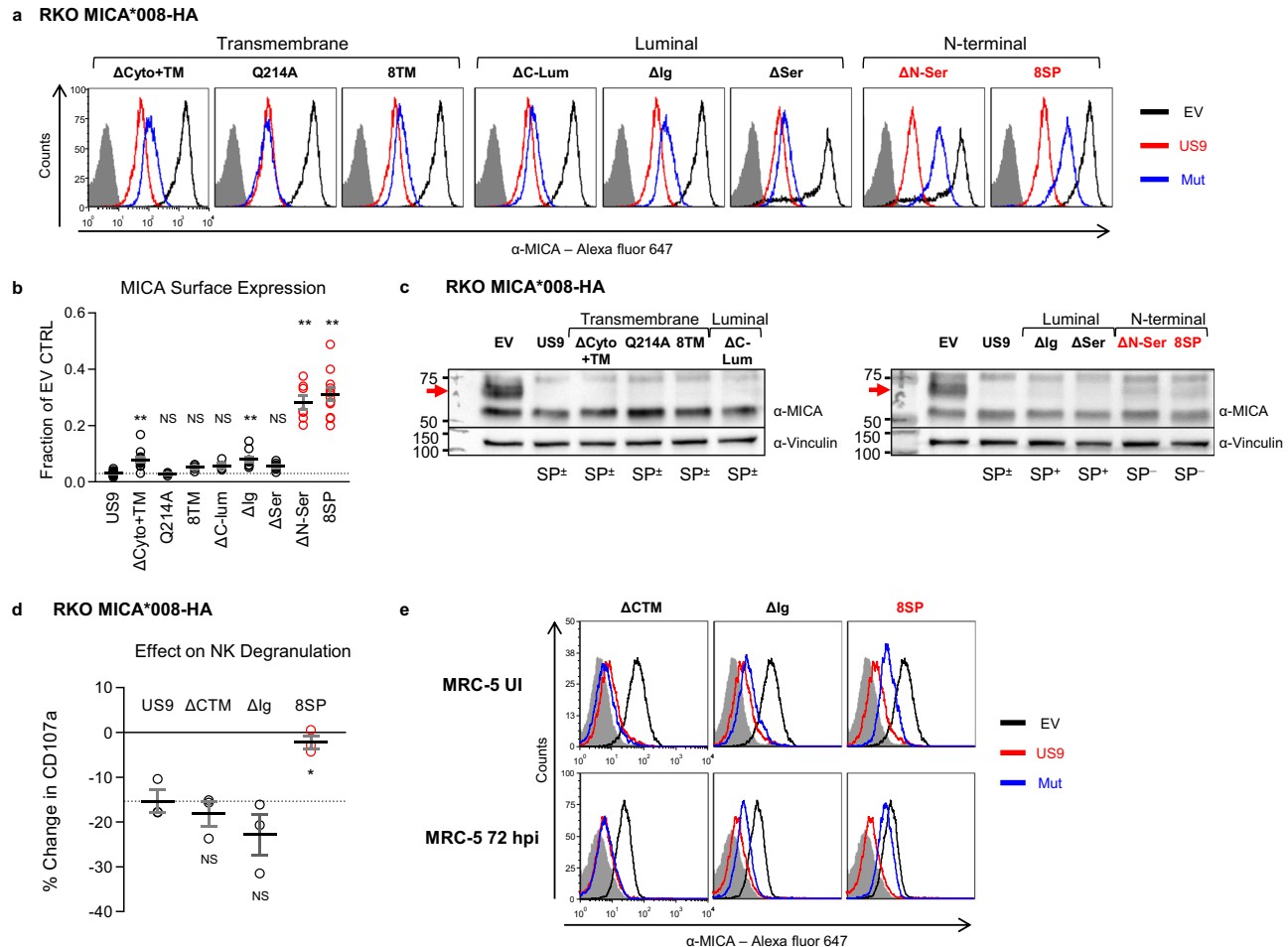

**Fig. 3 US9 N-terminus is required for effective MICA*008 downregulation and immune evasion. a** Flow cytometry of MICA surface expression in RKO MICA*008-HA cells expressing an EV (black), US9 (red), or the indicated US9 mutants (blue), grouped by mutated region (indicated). Red font highlights MICA*008 downregulation impairment. Gray-filled histograms represent secondary antibody staining of EV cells. Figure combines representative results from separate experiments. **b** Quantification of MICA mean flourescence intensity (MFI) shown in (**a**), normalized to the EV control. Data show mean ± SEM for at least four independent experiments per mutant. A one-way ANOVA was performed with a significant effect at the $p < 0.05$ level for all conditions [$F(8,75) = 70.48$, $p = 9.2 \times 10^{-32}$]. MICA MFI for US9 (dashed line) was compared to each mutant's MFI using a post hoc Dunnett's test. \*\*$p < 0.01$. NS = non-significant. Red circles highlight MICA*008 downregulation impairment. **c** Lysates from the indicated cells described in (**a**) were blotted using anti-MICA, anti-His tag, and anti-vinculin as loading control. Panels depict separate gels. Annotation beneath the panels indicates SP status of the US9 constructs (SP$^{+/-/\pm}$). Red arrows indicate post-ER MICA*008. Red font highlights MICA*008 degradation impairment. Representative of three independent experiments. **d** NK cell degranulation following incubation with the indicated target cells was measured by CD107a staining. Relative degranulation reduction was calculated for each US9 mutant. Shown are mean ± SEM from three independent repeats. A one-way ANOVA was performed with a significant effect at the $p < 0.05$ level for all conditions [$F(3,8) = 8.52$, $p = 0.0071$]. US9 degranulation reduction (dashed line) was compared to each mutant using a post hoc Dunnett's test. \*$p < 0.05$. NS = non-significant. Red circles highlight degranulation reduction impairment. **e** The indicated MRC-5 fibroblasts were uninfected (UI; top panels) or infected with ΔUS9 HCMV and stained for MICA expression at 72 h post infection (hpi; bottom panels). Gray-filled histograms represent isotype-control staining of EV cells. Red font highlights MICA*008 downregulation impairment. Representative of three independent experiments. See also Table 1, Fig. S2, and the Source data file for experimental data and full statistics.

ΔN-Ser, and 8SP (Fig. 3a, b, highlighted in red). MICA*008 surface levels were increased by about tenfold in cells expressing these mutants compared to WT US9-expressing cells.

To determine if the mutants still induced MICA*008 degradation, we assessed their effect on total MICA*008 protein quantity by immunoblot. We have previously shown that US9 does not reduce the quantity of ER-resident, non-GPI-anchored MICA*008, while post-ER, GPI-anchored MICA*008 vanishes[28]. In RKO cells, the ER-resident and post-ER forms of MICA*008 are easily distinguishable by size since the post-ER form of MICA*008 migrates at ~70 kDa due to Golgi-acquired glycosylation modifications, compared to the ~60 kDa ER-resident form[28]. Most of the US9 mutants maintained this pattern (Fig. 3c), substantially reducing the quantity of post-ER

MICA*008 (red arrows) compared to empty-vector (EV) control cells. Only the two N-terminal SP⁻ mutations partially restored post-ER MICA*008 quantities (Fig. 3c, red font), demonstrating impaired MICA*008 degradation. In summary, we concluded that US9 operates differently than US2/11 and its main functional domain is localized to its N-terminus.

**US9 SP is required for evasion of NK cell-mediated killing.** We next wondered about the functional significance of the partial MICA*008 restoration in the N-terminal US9 mutants. To address this question, we conducted a CD107a-degranulation assay. CD107a is a cytotoxic granule marker transiently expressed on NK cell surface following degranulation[51]. NK cells were co-incubated for 2 h with target cells expressing EV, US9, ΔCyto+TM, ΔIg, or

**Table 1 Characteristics and function of US9 mutants.**

| Construct name | Mutated domain | US9 SP status | Effect on MICA*008[a] | Related figure |
|---|---|---|---|---|
| US9 | — | Slowly cleaved | ++ | Fig. 3 |
| ΔCyto+TM | Cytosolic and transmembrane | Slowly cleaved | ++ | |
| Q214A | Transmembrane | Slowly cleaved | ++ | |
| 8TM | Transmembrane | Slowly cleaved | ++ | |
| ΔC-Lum | Luminal | Slowly cleaved | ++ | |
| ΔIg | Luminal | Uncleaved | ++ | |
| ΔSer | Luminal | Uncleaved | ++ | |
| ΔN-Ser | N-terminal | Rapidly cleaved | + | |
| 8SP | N-terminal | Swapped | + | |
| 8SPΔCyto | N-terminal, Cytosolic | Swapped | + | Fig. 5 |
| 8SPΔCyto+TM | N-terminal, Cytosolic and transmembrane | Swapped | − | |
| 8SPQ214A | N-terminal, transmembrane | Swapped | + | |
| 8SP8TM | N-terminal, transmembrane | Swapped | + | |
| 8SPΔC-Lum | N-terminal, luminal | Swapped | + | |
| 8SPΔIg | N-terminal, luminal | Swapped | − | |
| 8SPΔSer | N-terminal, luminal | Swapped | + | |
| 8SPΔN-Ser | N-terminal | Swapped | + | |

[a]++: MICA*008 surface expression <10% of empty-vector (EV) control; +: MICA*008 surface expression 10–75% of EV control; −: MICA*008 surface expression >75% of EV control.

8SP. The percentage of CD107a+ NK cells was measured by flow cytometry, and the reduction in NK degranulation induced by US9 and the mutants was calculated (Fig. 3d). US9, ΔCyto+TM, and ΔIg, reduced NK cell degranulation by 15–20% compared to EV control and there was no significant difference between them. In contrast, 8SP was significantly less efficient than US9 and failed to reduce NK cell degranulation, indicating that SP removal hampered US9 NK cell evasion capacity.

**US9 SP is required for MICA*008 targeting during HCMV infection.** We wondered what the impact of the US9 SP deletion during HCMV infection would be, since its effect could potentially be masked by other, as-yet unidentified MICA*008-targeting viral genes[28]. To test this, we expressed EV, US9, ΔCyto+TM, ΔIg, or 8SP in HCMV-permissive MRC-5 fibroblasts, which we genotyped and determined to be MICA*008 homozygous. We then infected the transduced MRC-5 cells with an HCMV mutant lacking US9 (ΔUS9) we previously generated[28]. We assessed MICA*008 surface expression in uninfected (UI) and infected cells at 72 h post infection (hpi; Fig. 3e). In uninfected cells, only 8SP partially restored MICA*008 levels. Following infection, ΔCyto+TM behaved similarly to WT US9, ΔIg showed partial impairment in MICA*008 downregulation, and 8SP was greatly impaired in MICA*008 downregulation, to the extent that MICA*008 levels in 8SP-expressing cells approached those in EV control cells. We therefore concluded that the US9 SP was the most significant MICA*008-targeting domain of US9, even in the context of HCMV infection.

**US9 SP is sufficient for inducing MICA*008 degradation.** Substitution of the SP and deletion of the N-Ser domain both impaired US9 function to a similar extent. There are two possible explanations to this finding: that both sequences were required for N-terminal domain function; or that the SP is an independent functional domain, and the N-Ser area has a supporting role in delaying its cleavage. To address this and discover whether the SP was sufficient for MICA*008 downregulation, we generated reciprocal chimeras based on US8 (Fig. 4a). First, we replaced US8's endogenous SP with that of US9 alone (9SP) or together with the adjacent N-Ser domain (9SP+N-Ser). In addition, to discover if the N-Ser domain is independently functional, we

wanted to insert it alone into US8, but a SP was needed to ensure ER insertion. Therefore, we first had to show that the US9 N-terminal domain could downregulate MICA*008 even when inserted after a different SP. We inserted the US9 SP and the N-Ser area (a total of 40 AA) after the US8 endogenous SP as a control (40ins). We inserted an extended sequence including the last seven AA of the SP and the N-Ser area (a total of 20 AA) after US8's SP (20ins), to increase the chances that the inserted sequence would be functional.

We overexpressed the US8 chimeras in RKO MICA*008-HA cells and verified by immunoblot all were expressed (Fig. 4b). Intriguingly, we observed that the 9SP chimera was larger than WT US8. Since the only sequence difference between the two is the SP itself, this showed that the US9 SP was constitutively retained on this construct (SP+). The 9SP+N-Ser chimera regained the typical doublet appearance of US9, indicating that the addition of the N-Ser domain restored slow SP cleavage (SP±). The 40ins chimera looked identical to 9SP+N-Ser in forms and size, indicating that the US8 SP was rapidly cleaved, and then the US9 SP was slowly cleaved (SP±). In contrast, the 20ins chimera had only one form which was larger than the SP− form of 40ins/9SP + N-Ser, indicating the SP cleavage site in the extended N-Ser sequence we inserted was not functional and that the entire inserted sequence was retained (SP−). Since US8 is known to partially localize to the Trans-Golgi network and lysosomes, rendering it endoH resistant[31,49,50], we used Peptide:N-glycosidase F (PNGaseF), an enzyme that digests all N-linked glycans, to validate the US9 SP status of the chimeras (Fig. S3a, annotated beneath the panel).

We assessed the cellular localization of US8 and the chimeras by an endoH assay (Fig. S3b) and found that SP+ forms seemed to be entirely endoH-sensitive, indicating they were being retained in the ER. In comparison, 20ins gained partial endoH resistance similarly to WT US8.

We then measured the chimeras' effect on MICA*008 surface expression by flow cytometry (Fig. 4c, quantified in Fig. 4d). While parental US8 and the 20ins chimera had no effect on MICA*008 surface expression, 9SP, 9SP+N-Ser, and 40ins all robustly downregulated MICA*008 by ~10-fold. In agreement with the surface expression data, immunoblot analysis (Fig. 4e) showed that 9SP, 9SP+N-Ser, and 40ins greatly reduced the quantity of post-ER MICA*008 (red arrow), showing they, too,

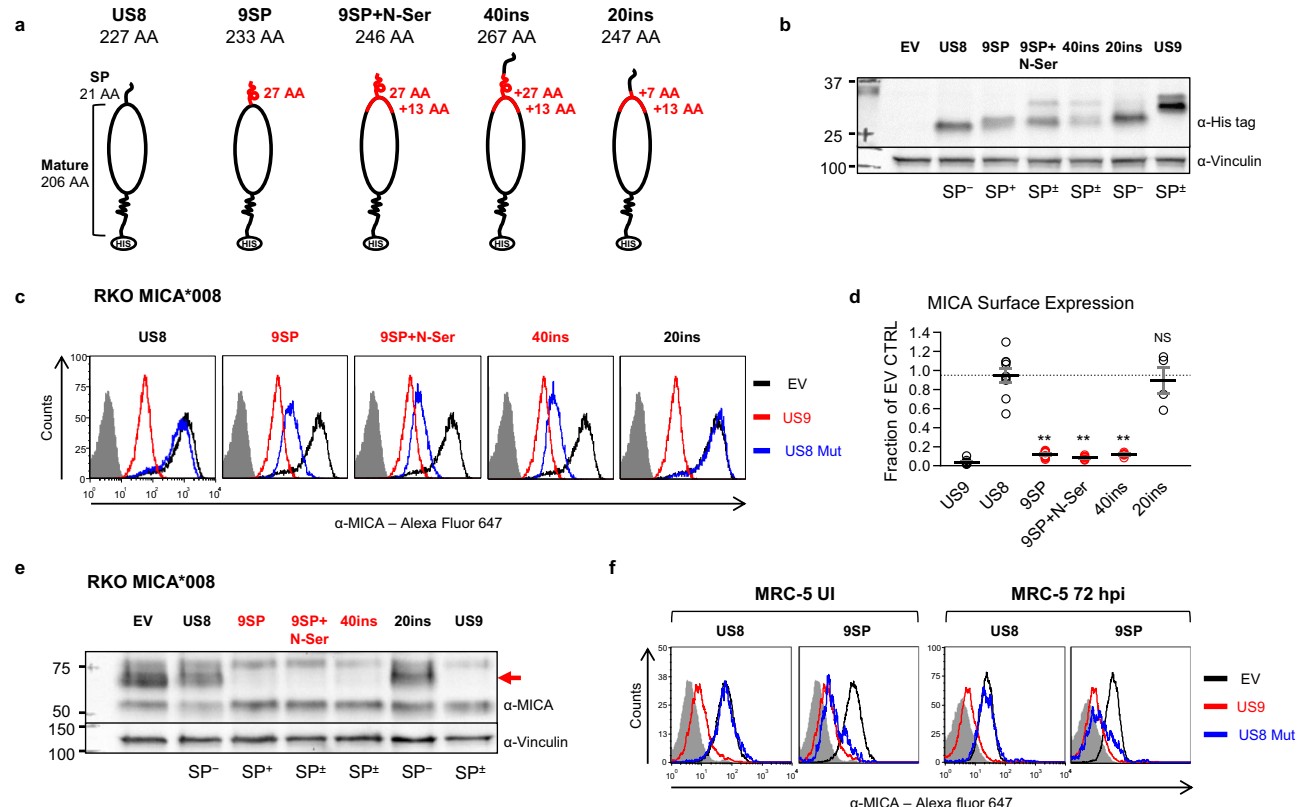

**Fig. 4 The US9 SP is sufficient for targeting MICA*008 to degradation. a** Schematic representation of US8 and the indicated US8 mutants. US9 domains of the indicated length (solid red lines) substituted US8's endogenous SP or were inserted after it. The total length of each construct without tags is indicated. **b** Lysates from RKO MICA*008-HA cells transduced with the constructs described in (**a**) were immunoblotted. Anti-His tag was used to visualize protein expression, with anti-vinculin as loading control. Annotation beneath the panel indicates the US9 SP status of the constructs ($SP^{+/-/\pm}$). Representative of three independent experiments. **c** Flow cytometry of MICA surface expression in cells transduced with an EV (black histograms) US9 (red histograms) or the indicated US8 constructs (blue histograms). Red font highlights gain-of-function constructs. Gray-filled histograms represent secondary antibody staining of EV cells. **d** Quantification of MICA MFIs shown in (**c**), normalized to the EV control. Data show mean ± SEM for at least four independent repeats per mutant. A one-way ANOVA was performed with a significant effect at the $p < 0.05$ level for all conditions [$F(4,31) = 68.528$, $p = 6 \times 10^{-15}$]. A post hoc Dunnett's test was used to compare the MFI for 8SP (dashed line) to that of each mutant. **$p < 0.01$, NS non-significant. Red circles highlight gain-of-function. **e** Lysates obtained from the indicated mutants described in (**a**) were blotted using anti-MICA, anti-His tag, and anti-vinculin as loading control. Red font highlights gain-of-function, and the red arrow indicates post-ER MICA*008. Representative of three independent experiments. **f** MRC-5 fibroblasts were transduced with an EV (black histograms), WT US9 (red histograms) or the indicated US8 constructs (blue histograms), and then left uninfected (UI; left panels) or infected with ΔUS9 HCMV and stained for MICA expression at 72 h post infection (hpi; right panels), as part of the same experiment shown in Fig. 3e. Gray-filled histograms represent isotype-control staining of EV cells. Red font highlights gain-of-function. Representative of three independent experiments. See also Figs. S3–4 and the Source data file for experimental data and full statistics.

induced its degradation. ER-resident MICA*008 forms were spared, recapturing the US9 phenotype. These results show that US9 SP is sufficient for targeting MICA*008, and that the N-Ser domain does not independently target MICA*008.

**US9 SP is sufficient for MICA*008 targeting during HCMV infection**. Since we observed that the US9 SP was sufficient for targeting MICA*008, we wondered whether the same would be true during HCMV infection. We overexpressed EV, US9, US8, and 9SP in MRC-5 fibroblasts, infected them with ΔUS9 HCMV, and measured MICA*008 surface expression at 72 hpi compared to uninfected (UI) cells (Fig. 4f). Both in UI and in ΔUS9-infected cells, the 9SP construct reduced MICA*008 levels to the same extent as US9 itself.

**US9 SP is functional when affixed to the unrelated protein EYFP**. To address the possibility that some structural feature conserved in US8 and US9 was required for US9 SP function, we

decided to attach it to an unrelated protein—enhanced yellow fluorescent protein (EYFP) (Fig. S4a). N-terminal tags were removed from parental FLAG-HA-tagged EYFP protein (named FLAG-HA-EYFP), and the following chimeras were generated: 8SP EYFP, 9SP EYFP, and 9SP+N-Ser EYFP.

We transfected the EYFP constructs into RKO MICA*008-HA cells and checked their expression by immunoblot (Fig. S4b). EYFP harbors no glycosylations so we did not conduct an endoH assay. We observed a doublet appearance of 9SP EYFP, indicating that the US9 SP sequence was sufficient for inducing slow cleavage in the context of the EYFP sequence. Importantly, the relative abundance of $SP^+$ EYFP increased with addition of the N-Ser area, in support of this sequence's role in slowing SP cleavage.

We then assessed the effect on MICA*008 surface expression by flow cytometry (Fig. S4c, quantified in Fig. S4d). While parental EYFP and 8SP EYFP did not downregulate MICA*008, 9SP EYFP moderately downregulated MICA*008 surface levels, and 9SP+N-Ser EYFP (both highlighted in red) strongly

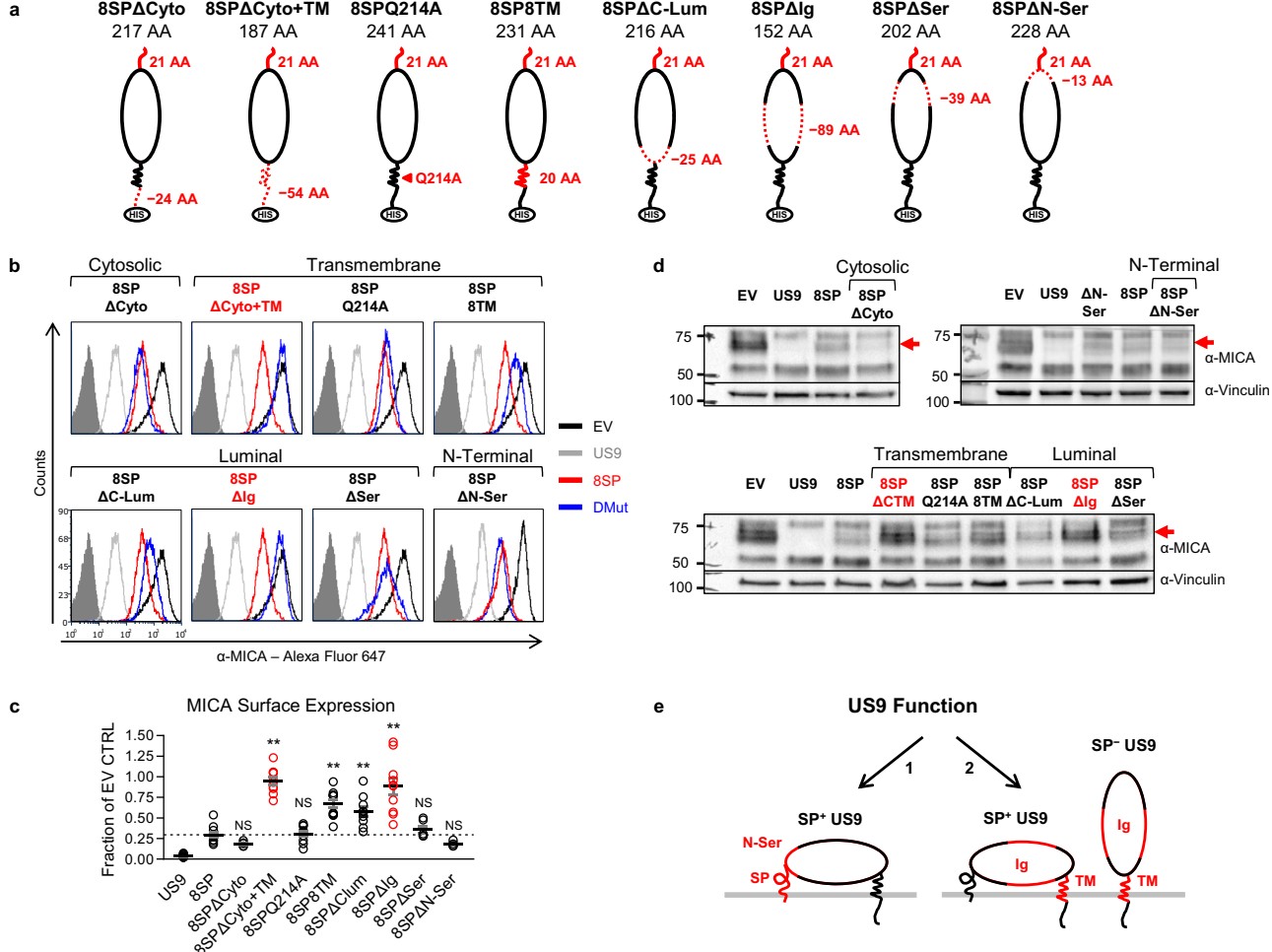

**Fig. 5 SP-independent MICA*008 downregulation is mediated by the US9 Ig-like and TM domains. a** Schematic representation of US9 double mutants, in which the SP was swapped with the US8 SP and combined with each of the indicated mutations in the following domains: serine-rich area (Ser), Ig-like fold (Ig), other luminal (Lum), transmembrane (TM), and cytosolic (Cyto). Dashed red lines show deleted domains, solid red lines show domains substituted with a homologous US8 sequence, arrow shows a single mutated residue. Mutants were 6XHis tagged (HIS). Mutated sequence length and the total length of each construct without tags are indicated. **b** Flow cytometry of MICA surface expression in RKO MICA*008-HA cells transduced with an EV (black histograms), US9 (gray histograms), 8SP (red histograms), or the indicated US9 double mutants (blue histograms). Gray-filled histograms represent secondary antibody staining of EV cells. Mutants are grouped by mutated region (indicated). Red font highlights complete loss-of-function. Figure combines representative results from separate experiments. **c** Quantification of MICA MFIs shown in (**b**), normalized to the EV control. Data show mean ± SEM for at least four independent experiments per mutant. A one-way ANOVA was performed with a significant effect at the $p < 0.05$ level for all conditions [$F(8,69) = 20.073$, $p = 2.8 \times 10^{-15}$]. A post hoc Dunnett's test was used to compare the MFI for 8SP (dashed line) to that of each mutant. **$p < 0.01$, NS non-significant. **d** Lysates obtained from the indicated cells were blotted using anti-MICA, anti-His tag, and anti-vinculin as loading control. Different panels show different gels. Mutants are grouped by mutated region (indicated). Red font highlights complete loss-of-function, and the red arrow indicates post-ER MICA*008. Representative of three independent experiments. **e** Diagram of the two distinct mechanisms of US9-mediated MICA*008 degradation: (1) US9 SP-dependent mechanism, present only in the SP+ isoform, requires the SP and the cleavage-delaying N-Ser domain; and (2) US9 SP-independent mechanism, present in both SP+ and SP− isoforms, requires the Ig-like and TM domains. See also Fig. S5, Table 1, and the Source data file for experimental data and full statistics.

downregulated MICA*008 surface levels to about 25% of the FLAG-HA-EYFP control levels. No other stress-induced ligand or MHC I were affected by the US9 SP, confirming that this effect was MICA-specific (Fig. S4e). We then tested total MICA*008 protein levels by immunoblot (Fig. S4f), to determine if the EYFP chimeras merely downregulated surface MICA*008 or caused its degradation. Expression of 9SP+N-Ser EYFP and to a lesser extent of 9SP EYFP (both highlighted in red), induced MICA*008 degradation since the post-ER form of MICA*008 (shown by a red arrow) was reduced. Importantly, the magnitude of the effect on MICA*008 correlated with SP+ EYFP abundance (Fig. S4d). These results demonstrate that US9 SP is sufficient for MICA*008 degradation when attached to EYFP.

**SP-independent US9 function is mediated by the Ig-like and TM domains**. The SP is the major functional domain of US9 in effect size and functional significance. However, SP− US9 still reduces MICA*008 levels by about 70% (Fig. 3b). To characterize the SP-independent functional domains, we constructed a panel of US9 double mutants. The 8SP mutation was combined with each of eight other mutations (Fig. 5a): deletion of the cytoplasmic domain (8SPΔCyto), or of the cytoplasmic and TM domains (8SPΔCyto+TM); mutation of a conserved TM glutamine to alanine (8SPQ214A); substitution with US8's TM domain (8SP8TM); deletion of the C-terminal part of the luminal domain up to the Ig-like fold (8SPΔC-lum); deletion of the Ig-like fold (8SPΔIg); deletion of the serine-rich area

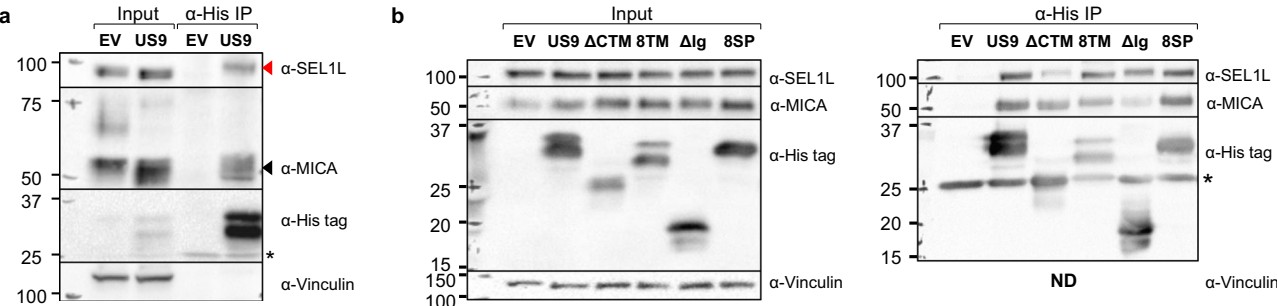

**Fig. 6 The US9 Ig-like and transmembrane domains, but not the US9 SP, bind MICA*008, and the ERAD scaffold protein SEL1L. a** Lysates prepared from RKO MICA*008-HA cells co-transduced with EV or US9 were immunoprecipitated using anti-His tag antibody. Immunoblotting was performed using anti-His tag, anti-MICA, and anti-SEL1L to visualize protein co-precipitation, with anti-vinculin as input loading control. Arrows indicate bands which specifically co-precipitated with US9 (red—SEL1L; black—MICA*008). * antibody light chain. Representative of three independent experiments. **b** Lysates prepared from the indicated RKO MICA*008-HA transfectants were immunoprecipitated using anti-His tag (A—input; B—IP) and immunoblotted using anti-His tag, anti-MICA, and anti-SEL1L. Anti-vinculin served as input loading control. * antibody light chain. Representative of two independent experiments. ND not detected. See also Fig. S6 and Supplementary Data 6.

(8SPΔSer); and last, deletion of the 13 N-terminal AA of the serine-rich area (8SPΔN-Ser).

The double mutants were then transduced into RKO MICA*008-HA cells, and their expression was confirmed by immunoblot (Fig. S5a). ER localization was confirmed by endoH digestion or by confocal microscopy co-localization for 8SPΔIg, which lacks N-linked glycosylations (Fig. S5b-c). As expected, following deglycosylation, all double mutants migrated as a single band consistent with rapid SP processing, and their sizes matched the expected sizes for SP⁻ forms.

We then measured MICA*008 surface expression by flow cytometry (Fig. 5b, quantified in Fig. 5c, summarized in Table 1). Strikingly, only two double mutants completely lost their capacity to downregulate MICA*008 surface levels: 8SPΔCyto+TM and 8SPΔIg, indicating that the TM and Ig two domains are required for SP-independent US9 function. All other double mutants retained their function either fully (8SPΔCyto, 8SPQ214A, 8SPΔSer, and 8SPΔN-Sser) or partially (8SP8TM and 8SPΔC-Lum). It is notable that the two N-terminal mutations, 8SP and ΔN-Ser were not additive, further evidence that they disrupt the same mechanism. Immunoblot analysis of total MICA*008 protein quantity in the presence of the US9 double mutants (Fig. 5d) confirmed that the 8SPΔCyto+TM and 8SPΔIg mutants do not reduce post-ER MICA*008 protein quantity (shown by a red arrow), while the other mutants remained capable of inducing its degradation, to a degree which correlated with MICA*008 surface expression. We therefore concluded that the US9 TM and Ig-like domains mediate SP-independent MICA*008 degradation, resembling US2/11.

In summary, we found that US9 contains two independent MICA*008-targeting mechanisms (Fig. 5e, functional domains highlighted in red): the first and most significant (1) is SP-mediated, present only in the SP⁺ isoform. This mechanism depends on the presence of the N-Ser domain which slows SP cleavage. The second, SP-independent mechanism (2) is mediated by the TM and Ig-like domains and is present in SP⁺ and SP⁻ isoforms.

**US9 co-precipitates with the ERAD component SEL1L and with MICA*008.** Having identified US9's functional domains, we wanted to study their underlying mechanisms of action. We began by performing an anti-His tag co-immunoprecipitation assay in EV controls and in US9-His-expressing cells to identify potential US9 interacting proteins. We used RKO MICA*008-HA cells and HeLa cells which endogenously express MICA*008[28]. Immunoprecipitated proteins were subsequently analyzed by LC-

MS/MS. We searched the results for proteins which specifically co-precipitated with US9 in both cell types. These analyses identified Protein Sel-1 Homolog 1 (SEL1L), an essential component of the HRD1 ERAD complex[52], as a US9 interacting protein (mass spectrometry data available in Supplementary Data 6). Hence, we chose SEL1L for further analysis.

First, we performed an anti-His tag IP and immunoblot to validate the interaction between US9 and SEL1L in RKO MICA*008-HA and in Hela cells (Figs. 6a and S6a, respectively). US9 specifically co-precipitated with SEL1L (shown by a red arrow). We also found that US9 co-precipitated with ER-resident MICA*008 (shown by a black arrow). To determine if the SEL1L-US9 interaction was MICA*008-dependent, we repeated the anti-His tag immunoprecipitation assay in RKO cells expressing MICA*004-HA, a full-length MICA allele which is not targeted by US9[28] (Fig. S6b). As expected, MICA*004 did not co-precipitate with US9, but SEL1L still co-precipitated with US9 (red arrow), demonstrating that its interaction with US9 does not require MICA*008.

**The US9 TM and Ig-like domains bind SEL1L and MICA*008, respectively.** Having established that US9 binds MICA*008 and SEL1L, we asked which US9 domains were responsible for these interactions. We repeated the co-immunoprecipitation described in Fig. 6a to test which US9 mutants would lose their ability to co-precipitate with ER-resident MICA*008 or with SEL1L. We tested the 8SP, ΔCyto+TM, 8TM, and ΔIg mutants (Fig. 6b). Strikingly, despite being functionally impaired, the 8SP mutant was still able to co-precipitate with both MICA*008 and SEL1L. The ΔIg mutant showed diminished MICA*008 binding but continued to interact with SEL1L. Conversely, the ΔCyto+TM mutant showed diminished SEL1L binding but continued to interact with MICA*008. It is intriguing that the 8TM mutant, in which the US9 TM domain was swapped with the equivalent US8 sequence, continued to bind SEL1L, suggesting that US8 itself might also bind this factor. These results are consistent with the requirement for the Ig-like and TM domains for MICA*008 targeting by SP⁻ US9 (Fig. 5). We concluded that the Ig-like domain binds MICA*008 and the TM domain binds SEL1L, while the SP does not bind either protein.

**US9-mediated degradation of MICA*008 requires SEL1L.** Based on these IP results, we hypothesized that only SP-independent US9 function requires SEL1L. To investigate the functional significance of the US9-SEL1L interaction, we knocked down SEL1L. RKO MICA*008-HA cells were transduced with a scrambled

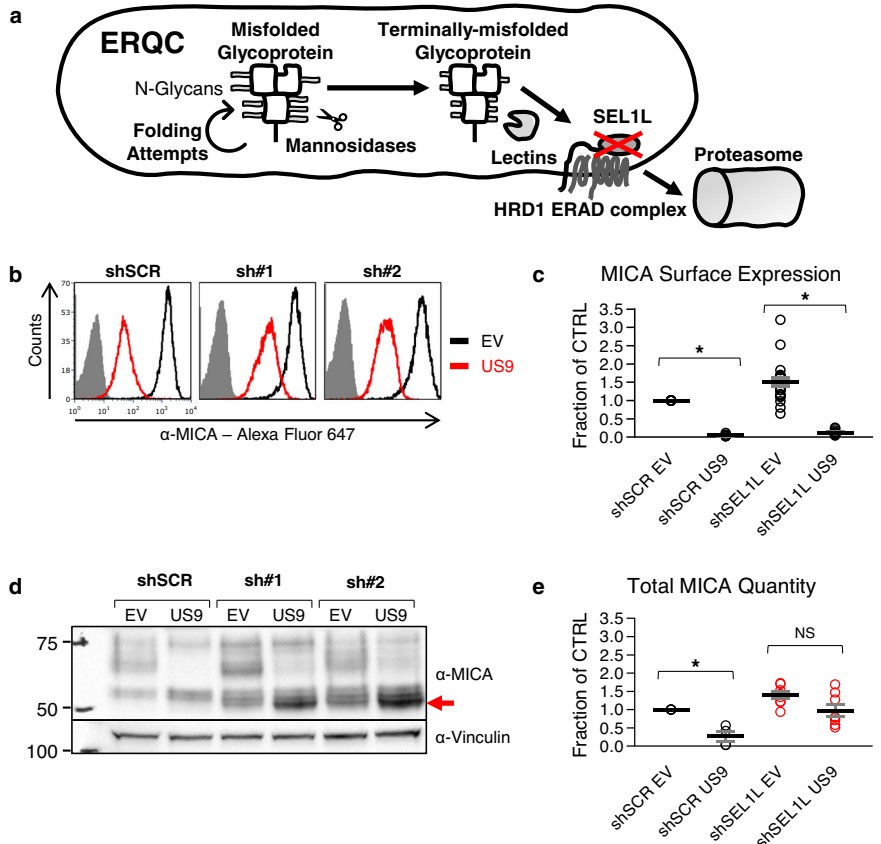

**Fig. 7 SEL1L is required for US9-mediated MICA*008 degradation. a** Schematic representation of the effect of SEL1L knockdown (red X) on the ER quality control (ERQC) pathway. Misfolded glycoproteins undergo folding attempts while mannosidases progressively trim their N-glycans, until lectins deliver terminally trimmed glycoproteins to the SEL1L-HRD1 ER-associated degradation (ERAD) complex, which retrotranslocates them to the proteasome. Therefore, SEL1L knockdown should induce the accumulation of terminally trimmed misfolded glycoproteins. **b** RKO MICA*008-HA cells transduced with the indicated shRNAs (shScrambled, shSEL1L#1 and shSEL1L#2, annotated as shSCR, sh#1 and sh#2), were co-transduced with an EV or with US9 and stained for MICA surface expression. Gray-filled histograms represent secondary antibody staining of EV cells. **c** Quantification of MICA MFIs shown in (**b**). MFIs were normalized to the MFI of shSCR EV cells. Data show mean ± SEM for eleven independent experiments. Two-tailed unequal variance Student's $t$-tests were used to compare EV vs. US9 surface MICA levels in shSCR cells and in pooled shSEL1L cells (shSCR $p = 1.92 \times 10^{-17}$, shSEL1L $p = 9.63 \times 10^{-11}$). Sidak's correction was used to determine a corrected alpha level, *$p < 0.0252$. **d** Lysates obtained from the indicated cells were blotted using anti-MICA and anti-vinculin loading control. The red arrow indicates glycosylation-trimmed MICA*008. **e** Total MICA*008 protein levels shown in (**d**) were quantified relative to the shSCR EV control and normalized to the loading control. Data show mean ± SEM for four independent experiments. Two-tailed unequal variance Student's $t$-tests were used to compare EV vs. US9 total MICA levels in shSCR cells and in pooled shSEL1L cells (shSCR $p = 0.012$, shSEL1L $p = 0.043$). Sidak's correction was used to determine a corrected alpha level, *$p < 0.0252$. NS = not significant. See also Figs. S7–8 and the Source data file for experimental data.

shRNA control (referred to as shSCR) or with two different shRNAs directed at SEL1L: shRNA#1 and shRNA#2 (referred to as sh#1 and sh#2, respectively). Once stably expressing the shRNAs, cells were co-transduced with an EV control or with US9. Knockdown (KD) efficiency was validated by immunoblot (Fig. S7a).

The SEL1L-HRD1 ERAD complex degrades misfolded luminal and transmembrane glycoproteins, including MHC I. Terminal trimming of N-linked glycans on the misfolded substrates causes dissociation from folding factors and recognition by lectins, which deliver the misfolded glycoprotein to the SEL1L-HRD1 ERAD complex for retrotranslocation and subsequent proteasomal degradation[9,52–55]. In addition to its role in quality control, the SEL1L-HRD1 complex regulates the quantity of folding-competent proteins, and is essential for many physiological and homeostatic functions, including energy metabolism, water balance, and cellular development[56–62]. Therefore, SEL1L knockdown is expected to induce accumulation of terminally trimmed misfolded glycoproteins (Fig. 7a).

We stained the shRNA-expressing cells for surface MICA*008 expression (Fig. 7b, quantified in Fig. 7c). SEL1L KD somewhat increased MICA surface levels in EV and US9-expressing cells, but US9 still significantly reduced MICA*008 surface levels in the SEL1LKD cells by about tenfold (Fig. 7c). Next, we immuno-blotted lysates from the shRNA-expressing cells (Fig. 7d, quantified in Fig. 7e). In the SEL1L KD EV cells, surface and total MICA*008 levels increased by about 40–50% compared to shSCR EV cells, implying a role for SEL1L in intrinsic MICA*008 degradation. Importantly, even though SEL1L KD did not restore surface MICA*008, total MICA*008 protein levels were greatly increased in the KD cells compared to shSCR US9 cells, due to accumulation of a rapidly migrating MICA*008 form (shown by a red arrow). Moreover, quantitation revealed no significant differences in total MICA*008 protein levels when comparing EV to US9-expressing SEL1L KD cells (Fig. 7e). This implied that in the KD cells, MICA*008 was retained intracellularly and was no longer degraded by US9.

Intriguingly, the rapidly migrating MICA*008 form which accumulated in SEL1L KD US9 cells was smaller in size than MICA*008 in shSCR US9 cells (Fig. 7d, shown by a red arrow). This size difference vanished upon endoH treatment (Fig. S7b), revealing that the retained MICA*008 form was identical to the ER-resident, non-GPI-anchored, 37 kDa form found in shSCR US9 cells. Reduced glycosylation size reflected glycosylation trimming of the ER-retained MICA*008.

To directly assess the kinetics of MICA*008 maturation and degradation in US9-expressing shSCR and shSEL1L cells, we conducted a CHX chase assay. Cells were treated with translation inhibitor CHX and were lysed at different time points during a 5-h chase. Lysates were then immunoblotted to assess MICA*008 levels (Fig. S7c, ER-resident MICA*008 quantified in Fig. S7d). In US9 shSCR control cells, ER-resident MICA*008 quantity (red arrow) slowly declined but did not become mature MICA*008, indicating MICA*008 is lost primarily through degradation. Importantly, in the SEL1L KD cells, ER-resident MICA*008 quantity changed very little throughout the chase, showing that in US9-expressing SEL1L-deficient cells, MICA*008 is retained in the ER instead of being degraded. Using immunofluorescence, we visualized MICA*008 accumulation in SEL1L-deficient US9-expressing cells, and verified that retained MICA*008 co-localized with the ER marker PDI (Fig. S7e). In summary, in the absence of SEL1L, US9 failed to induce MICA*008 degradation. Instead, non-GPI-anchored, glycan-trimmed MICA*008 accumulated in the ER.

Since US9 acts as an ER-retention factor in the absence of SEL1L, we postulated that SEL1L KD would also affect MICA*008-US9 complex stability. To examine this, we performed an anti-HA tag co-immunoprecipitation in the RKO MICA*008-HA shSCR and shSEL1L cells (Fig. S7f). While US9 failed to co-precipitate with MICA*008 in the shSCR cells, it was recovered from the SEL1L KD cells (red arrows). We concluded that SEL1L KD led to the stabilization of the MICA*008-US9 complex.

In conclusion, we found that US9 lost its ability to degrade MICA*008 in SEL1L-depleted cells, even though only SP-independent functional domains directly bound SEL1L. This suggests that both SP-dependent and SP-independent US9 mechanisms require SEL1L for MICA*008 degradation.

**US9 SP arrests MICA*008 maturation despite lack of direct binding**. To directly assess the reliance of each US9 functional domain on SEL1L and establish which US9 domains mediate MICA*008 maturation arrest, we tested the impact of SEL1L KD on different US9 mutations. RKO MICA*008-HA expressing shSCR or shSEL1L#2 were co-transduced with EV, US9 or the following mutants: ΔIg which lacks MICA*008 binding; 8SP which lacks the US9 SP; and 9SP, in which only the US9 SP is present. We stained the cells for surface MICA*008 expression and quantified the results (Fig. 8a). SEL1L KD slightly increased MICA*008 surface levels across all cell types but did not restore MICA*008 surface levels in any of the US9 mutants. We then blotted lysates obtained from the various shSCR and shSEL1L#2 transfectants to visualize ER accumulation of MICA*008 (Fig. 8b). US9 and all mutants caused at least some degree of MICA*008 ER retention in the SEL1L KD cells, as shown by accumulation of the glycan-trimmed form (red arrow). This accumulation confirms that MICA*008 degradation was impaired in all US9 mutants in SEL1L's absence.

MICA*008 accumulation was very prominent in SEL1L KD cells expressing 9SP and ΔIg, two mutants where the US9 SP is the only active domain. 8SP, where the Ig-like domain is the only active one (since SEL1L is depleted so the TM domain cannot bind it), also retained MICA*008, but less robustly. We concluded that despite the lack of direct interaction between it and MICA*008, the US9 SP is the main mediator of MICA*008 maturation arrest in SEL1L-depleted cells, with a secondary contribution from the Ig-like domain.

**MICA*008 accumulates in a soluble form in SEL1L-depleted cells**. In addition to its role in ERQC as an indispensable part of the HRD1 E3-ligase complex, SEL1L also has HRD1-independent functions. In particular, SEL1L inhibits the aggregation of lipo-protein lipase (LPL) in the ER, enabling its maturation and ER egress. In SEL1L's absence, ER-retained detergent-insoluble LPL aggregates are degraded primarily by autophagy[56]. We therefore wondered whether SEL1L's effect is mediated by the HRD1 ubiquitin-proteasome pathway, or by another, independent mechanism.

First, we wanted to further characterize MICA*008 which accumulates in US9-expressing SEL1L-depleted cells. In immunofluorescence, ER-retained MICA*008 appeared diffusely distributed within the ER (Fig. S7e), rather than forming puncta that would suggest aggregation[56,58]. We next separated shSCR and SEL1LKD RKO MICA*008-HA cells expressing either EV or US9 into detergent-soluble and detergent-insoluble fractions (Fig. S7g). Mature MICA*008 is GPI-anchored and localizes to detergent-resistant membranes[22]. Accordingly, this form, present only in the EV-expressing cells, partitioned into detergent-insoluble pellets. In contrast, the ER-resident form of MICA*008 (red arrow) was detergent-soluble in EV and US9-expressing control cells. Importantly, in SEL1LKD cells, the ER-resident form remained largely soluble, even in the presence of US9, and only a minor quantity partitioned to the detergent-insoluble pellet. These results indicate that ER-retained MICA*008 does not require SEL1L to remain soluble. These results are supported by our previous finding that proteasomal but not lysosomal inhibitors impede US9-mediated degradation of MICA*008[28], ruling out autophagy as a mechanism of MICA*008 degradation.

**US9-mediated degradation of MICA*008 requires HRD1**. To directly address HRD1's function in MICA*008 degradation, we knocked down HRD1. We hypothesized that HRD1 KD would phenocopy SEL1L KD, and induce the accumulation of glycosylation-trimmed MICA*008 (Fig. S8a). RKO MICA*008-HA cells were transduced with a shSCR control or with two different shRNAs directed at HRD1: shRNA#3 and shRNA#4 (referred to as sh#3 and sh#4, respectively). Knockdown efficacy was verified by immunoblot (Fig. S8b). We then assayed MICA*008 surface expression by flow cytometry (Fig. S8c, quantified in Fig. S8d). Similar to SEL1L KD, HRD1 depletion mildly increased MICA*008 surface levels, but US9 still robustly reduced surface MICA*008 in the KD cells. Next, we assayed whole-cell MICA*008 levels by immunoblot (Fig. S8e, quantified in Fig. S8f). Again, as seen in SEL1L KD, depletion of HRD1 mildly increased total MICA*008 levels in the absence of US9, suggesting the SEL1L-HRD1 complex intrinsically degrades MICA*008. Crucially, HRD1 depletion resulted in the substantial accumulation of a rapidly migrating MICA*008 form (Fig. S8e, shown by a red arrow), and abolished the differences in total MICA*008 protein levels when comparing EV to US9-expressing HRD1 KD cells (Fig. S8f). This demonstrates that HRD1 KD impairs US9-mediated MICA*008 degradation, resembling SEL1L KD (Fig. 7d, e).

Last, to directly demonstrate that MICA*008 is degraded by the E3-ligase activity of HRD1, we immunoprecipitated MICA*008 from US9-expressing shSCR and shHRD1#4 RKO MICA*008-HA cells, using an anti-HA-tag antibody. Cells were treated with 4 μM of the proteasomal inhibitor EPX for 5 h prior

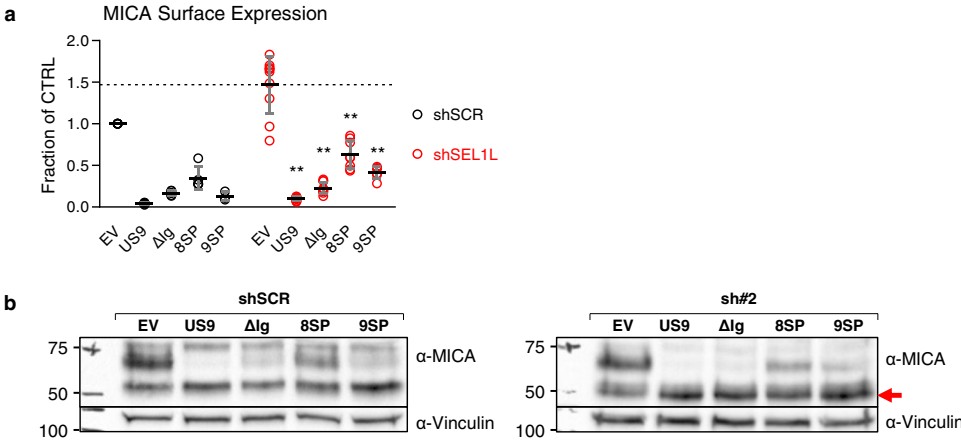

**Fig. 8 US9 SP arrests MICA*008 maturation in SEL1L's absence.** RKO MICA*008-HA cells expressing shScrambled (shSCR) or shSEL1L#2 (sh#2) were co-transduced the indicated constructs. **a** Quantification of MICA*008 surface expression on the indicated cells measured by flow cytometry. MFIs were normalized to the MFI of shSCR EV transfectants. Data show mean ± SEM for at least three independent experiments per construct. A two-way ANOVA (knockdown, US9 construct expression) was conducted to compare the normalized MFIs of the various constructs in shSCR vs. shSEL1L cells. The main effect of knockdown was significant at the $p < 0.05$ level [$F_{(1,56)} = 41.46$, $p = 2.01 \times 10^{-8}$]. The main effect of US9 construct expression was also significant [$F_{(4,56)} = 276.73$, $p = 3.59 \times 10^{-36}$]. There was a significant interaction between knockdown and US9 construct expression [$F_{(4,56)} = 20.16$, $p = 2.5 \times 10^{-10}$]. A post hoc Tukey test was used to compare EV MFI (dashed line) to that of each construct in the shSEL1L-expressing cells. **$p < 0.01$. **b** Lysates were obtained from the indicated cells and blotted using anti-MICA and anti-vinculin loading control. The red arrow indicates glycosylation-trimmed MICA*008. Representative of two independent repeats. See also Figs. S7–8 and the Source Data file for experimental data and full statistics.

to precipitation to induce the accumulation of polyubiquitiny-lated proteins. We immunoblotted the eluates using anti-MICA and anti-ubiquitin antibodies (Fig. S8g, relative quantification in Fig. S8h). While MICA*008 quantities increased in the HRD1 KD US9-expressing cells, ubiquitin-conjugated MICA*008 quantity decreased, so that the ratio of polyubiquitinylated MICA*008 to MICA*008 decreased by about 70%. Overall, these results show that MICA*008 is a bona fide substrate of the SEL1L-HRD1 ERAD complex.

**US9 SP harnesses physiological ERQC to induce MICA*008 degradation.** We found that the US9 SP induced MICA*008 degradation via the SEL1L-HRD1 complex without directly binding either, and that in SEL1L's absence, the SP retained MICA*008 in a terminally trimmed form. We therefore specu-lated that US9 SP might be blocking MICA*008 maturation, leading to progressive glycosylation trimming and eventual degradation by the endogenous ER quality control machinery.

To test this hypothesis, we used kifunensine (KIF), a specific inhibitor of α1,2 mannosidases which inhibits N-linked glycan trimming and is a potent ERAD inhibitor[63]. Treatment with KIF causes the accumulation of untrimmed misfolded glycoproteins (Fig. 9a). KIF also prevents Golgi-based glycosylation trimming and modification, so even post-ER glycoproteins remain endoH-sensitive[64].

We incubated RKO MICA*008-HA cells expressing EV or US9 with 200 μM KIF for 24 h and lysed the cells. To allow comparison of all different MICA*008 forms, lysates were either mock-treated or digested with endoH or PNGaseF, and immunoblotted (Fig. 9b; different MICA*008 forms are anno-tated). As expected, KIF treatment in the EV cells caused most of MICA*008 to lose its complex glycosylation modifications and consequently its size decreased (compare untreated and KIF-treated undigested EV samples). These non-modified glycosylations remained endoH-sensitive (compare KIF-treated EV samples digested with endoH to EV samples digested with PNGaseF). PNGaseF treatment revealed a band of ~42 kDa, which represents the deglycosylated native full-length

MICA*007:01 allele expressed by RKO cells at low levels, and this form (marked by *) also became endoH-sensitive following KIF treatment. In contrast, in US9-expressing cells, the size of glycosylated ER-resident MICA*008 increased, as expected due to the inhibition of ER glycosylation trimming (compare undigested US9 samples with and without KIF treatment).

Importantly, despite these differences in glycosylation size, KIF did not affect the GPI-anchoring status of MICA*008. In the EV cells, most MICA*008 remained in the GPI-anchored 34 kDa form upon deglycosylation. In the US9-expressing cells, the endoH-sensitive, non-GPI-anchored 37 kDa form is the only one present, and KIF treatment did not change this.

To compare the effect of KIF treatment on total MICA*008 quantity, PNGaseF-digested MICA*008, marked by red squares, was quantitated and the fold change in MICA*008 levels following KIF treatment was calculated (Fig. 9c). KIF treatment increased MICA*008 levels in EV cells by ~30%, but in US9-expressing cells, MICA*008 levels showed a 2.5-fold increase, indicating KIF treatment increased MICA*008 to significantly higher levels in the presence of US9.

Next, we wanted to test the impact of KIF treatment on US9 mutants which have or lack the SP. RKO MICA*008-HA cells expressing EV, US9, 8SP (that lacks the US9 SP), US8, and 9SP (that has only the US9 SP) were mock-treated or treated with KIF, lysed, and blotted as before. All lysates were digested with PNGaseF to facilitate quantitation (Fig. 9d), and the fold change in MICA*008 levels following KIF treatment was calculated (Fig. 9e). Strikingly, Only US9 and 9SP showed a significant increase of at least twofold in MICA*008 levels. EV, 8SP, and US8 all showed a mild increase of MICA*008 levels and were not statistically different from each other. These results show that only SP-dependent degradation of MICA*008 requires glycosyla-tion trimming, pointing to an indirect mechanism that harnesses physiological ER quality control.

## Discussion

The mechanistic understanding of the ways by which HCMV manipulates host cells has been instrumental in elucidating basic

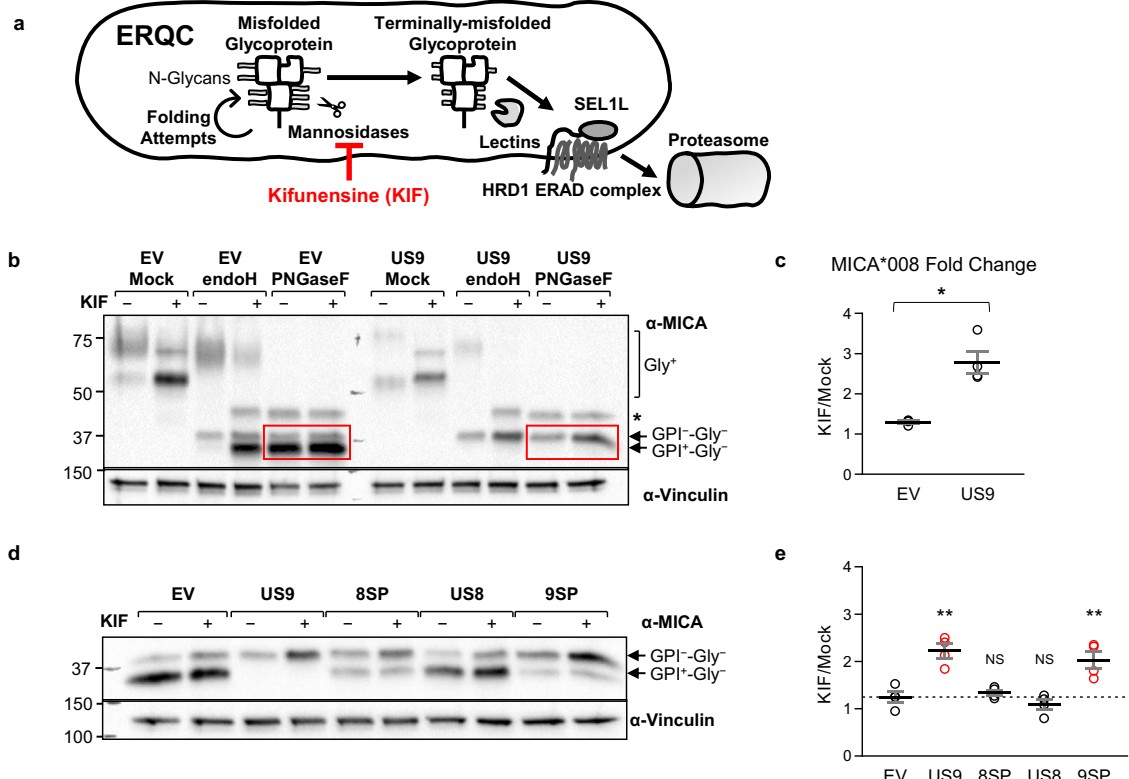

**Fig. 9 US9 SP arrests MICA\*008 maturation, indirectly leading to ERQC-mediated degradation. a** Schematic representation of the effect of kifunensine (KIF) treatment (red line) on the ER quality control (ERQC) pathway. KIF inhibits mannosidases, preventing glycosylation trimming of misfolded glycoproteins and subsequent ER-associated degradation (ERAD). Therefore, KIF treatment should cause the accumulation of untrimmed misfolded glycoproteins. **b** RKO MICA\*008-HA cells expressing EV or US9 were untreated (−) or treated (+) with 200 μM KIF and lysed. Lysates were mock-treated (Mock), or digested with endoH or with PNGaseF, and blotted using anti-MICA, with anti-vinculin as loading control. Annotated are the glycosylated forms of MICA\*008 (Gly$^+$), and deglycosylated MICA\*008 forms with or without the GPI anchor (GPI$^+$-Gly$^-$ and GPI$^-$-Gly$^-$, respectively). * Endogenous MICA\*007:01 (42 kDa). Red squares highlight PNGaseF-digested forms used for protein quantitation. **c** Deglycosylated MICA\*008 levels were normalized to the loading control and quantified relative to the mock EV control. The fold change in levels following KIF treatment was calculated. Data show mean ± SEM for four independent experiments. A two-tailed unequal variance Student's $t$-test was used to compare the fold change in EV vs. US9-expressing cells ($p = 0.011$). *$p < 0.05$. **d** The indicated cells were untreated (−) or treated (+) with KIF as described. Lysates were obtained, digested with PNGaseF, and blotted using anti-MICA, with anti-vinculin as loading control. Annotated are GPI-anchored (GPI$^+$-Gly$^-$) and unanchored (GPI$^-$-Gly$^-$) deglycosylated MICA\*008 forms. **e** MICA\*008 levels shown in (**d**) were normalized to the loading control and quantified relative to the untreated EV control. The fold change in total MICA\*008 protein levels following KIF treatment was calculated. Data show mean ± SEM for four independent experiments. A one-way ANOVA was performed with a significant effect at the $p < 0.05$ level for all conditions [$F(4,15) = 15.36$, $p = 3.4 \times 10^{-5}$]. A post hoc Dunnett's test was used to compare the fold change for EV (dashed line) to that of US9 and the mutants. Red columns show significant KIF-induced increase in total MICA\*008 levels. **$p < 0.01$, NS non-significant. See also the Source data file for experimental data and full statistics.

cellular processes. One such example is mammalian ERAD, where many of the key principles and components were discovered in a series of landmark studies on the US2/11-mediated degradation of MHC I[38,39,41,44,45,65,66]. We initially assumed that US9, which is structurally related to US2/US11, operates similarly. Instead, we found that US9 targets MICA\*008 to degradation by two distinct mechanisms: the dominant mechanism is SP-dependent, and only the second, accessory mechanism resembles US2/US11 function.

We found that like the US11 SP[32], the US9 SP is slowly cleaved, giving rise to two isoforms: SP$^+$ and SP$^-$. The SP sequence itself and the 13 AA flanking the cleavage site (N-Ser area) were both required for delayed SP cleavage in US9. However, cleavage kinetics were sensitive to additional sequence features. Large luminal deletions abrogated SP cleavage, possibly due to steric or conformational effects. Interestingly, when attached to other proteins, the SP alone could induce slow cleavage (EYFP) or entirely prevent cleavage (US8), indicating that the N-Ser area is dispensable in certain sequence contexts. Nonetheless, addition of N-Ser area always resulted in delayed SP cleavage.

Delayed cleavage permitted the SP to act first as an ER insertion signal, then as a transient transmembrane immune evasion domain. Rapid cleavage abolished SP immune evasion capacity, suggesting the SP cannot act as a free peptide. The reason for this requirement is not yet known. It may be the SP is degraded following its cleavage, or that cleavage otherwise inactivates it. Either way, when attached to any carrier protein, the SP alone was sufficient for MICA\*008 degradation provided it was uncleaved or slowly cleaved. This mode of SP function differs from previously described SP roles.

SPs are known to regulate ER insertion efficacy, folding steps within the ER, and even post-ER trafficking[5]. For instance, the SP of the HIV protein gp160 operates as an intramolecular chaperone: its cleavage is delayed until after proper protein folding, ensuring that only functional gp160 exits the ER[66–69]. Following their cleavage, SPs are generally degraded, but there are exceptions to this rule[3,5,7]. The best known example is that of the HCMV protein UL40, whose SP contains a conserved sequence also found in MHC I heavy chain SPs[70–73]. MHC I SP fragments are loaded onto the non-classical HLA-E molecule, an inhibitory NK ligand which serves as a reporter of proper MHC I expression.

UL40 SP mimicry preserves HLA-E surface expression even when other HCMV inhibitors target the MHC I pathway, thus eluding NK cell detection[74]. Cellular examples of SP cleavage controlling trafficking also exist, such as interleukin 15[75].

Notably, all these examples of SP functions are either regulatory, or they occur post-cleavage. We know of no other example where a SP acts as a protein-integral immune evasion domain prior to its cleavage, as shown here for US9.

Strikingly, SP-dependent downregulation of MICA*008 was the dominant US9 mechanism, accounting for a tenfold decrease in surface expression in a MICA*008/US9 overexpression model. Moreover, only SP-deficient US9 mutants were impaired in their ability to evade NK cell-mediated killing in this model. The SP also accounted for most of the reduction in surface MICA*008 in MRC-5 primary fibroblasts in an endogenous MICA*008/US9 overexpression model. Significantly, the same was true during infection of the MRC-5 fibroblasts with US9-deficient HCMV, showing that the SP effect was central even in the presence of other MICA*008-targeting viral mechanisms. An important caveat to this observation is that reconstitution by overexpression may not fully capture HCMV infection dynamics, since the US9 constructs were not under the control of the US9 promoter, but under the powerful immediate early CMV promoter. However, it is likely that if anything, this model overstated the effect of the SP-deficient mutants due to their high expression levels.

We found that US9 employs a second, SP-independent mechanism targeting MICA*008. In comparison to the potent SP-mediated effect, the SP-independent mechanism caused a threefold decrease in MICA*008 surface levels and failed to significantly affect NK-mediated killing in an overexpression model, and during HCMV infection it reduced MICA*008 expression only mildly. Mechanistically, the SP-independent US9 mechanism parallels the US2/US11 paradigm. The luminal Ig-like fold engages the degradation substrate, in this case MICA*008, and the TM domain recruits the ERAD complex, in this case the SEL1L-containing complex.

SEL1L is the scaffold upon which the HRD1 E3-ligase complex is assembled, interacting both with luminal ER chaperones that bind the degradation substrate and with the membrane-bound retrotranslocation machinery[10,52,76,77]. In the context of HCMV infection, SEL1L was initially implicated in US11 function[53] and later found to be essential for the degradation of unbound but not MHC-bound US11[48,65]. SEL1L was also found to mediate the degradation of misfolded secretory pathway substrates, among them MHC I[54].

Here, we showed that SEL1L knockdown completely abolished US9-mediated degradation of MICA*008. Both US9 mechanisms required SEL1L, though only the SP-independent domains bound SEL1L or MICA*008. Intriguingly, in SEL1L's absence, US9 caused MICA*008 to be retained in the ER in a non-GPI-anchored and glycosylation-trimmed form. The US9 SP was the main mediator of MICA*008 maturation arrest in SEL1L-depleted cells, with a secondary contribution from the Ig-like domain of the SP-independent mechanism. We also showed that HRD1 knockdown phenocopied the effects of SEL1L knockdown and confirmed that MICA*008 is a bona fide substrate of the SEL1L-HRD1 ERAD complex.

Finally, to address how the SP induced SEL1L-mediated degradation of MICA*008 without directly interacting with either protein, we showed that the SP relied on physiological ER quality control to "do its dirty work". Blocking glycosylation trimming, an essential step in the generation of degrons that mark terminally misfolded proteins to SEL1L-mediated degradation, prevented SP-dependent degradation of MICA*008, but had no effect on SP-independent degradation.

Based on these findings, we propose the following unified model for US9 function (Fig. 10a–c): newly synthesized non-GPI-anchored MICA*008 is recognized as misfolded and fails to pass ERQC checkpoints. MICA*008 is retained in the ER, undergoing repeated folding attempts and glycosylation trimming, until unknown factors rescue most of it by facilitating its GPI anchoring. GPI-anchored MICA*008 is then recognized as properly folded and can exit the ER toward the cell surface (Fig. 10a). US9 is synthesized in a SP$^+$ form and lingers in it, thanks to delayed SP cleavage. This permits the SP to interfere with unknown component(s) of MICA*008's non-standard maturation pathway. We speculate that these component(s) are also membrane-bound and thus can interact with the SP. As a result of this interference, non-anchored MICA*008 is trapped in the ER, undergoing progressive glycosylation trimming which eventually culminates in SEL1L-HRD1-mediated retrotranslocation and proteasomal degradation (Fig. 10b). In parallel, both the SP$^+$ and SP$^-$ forms of US9 deploy another, SP-independent mechanism which directly recruits SEL1L to MICA*008, bypassing the need for glycosylation trimming and recognition by lectins (Fig. 10c). This model can at least partially explain the slow kinetics of US9-induced MICA*008 degradation, as well as US9's dependence on the non-canonical GPI-anchoring pathway. In conclusion, the two US9 mechanisms act independently of each other and through different means yet share a common target—two immunoevasins for the coding space of one.

Several questions remain unresolved. First, we have not determined why SP-independent US9 degradation kinetics are slow, while US2/11 degrade MHC I within minutes[38,39]. It could be that interaction between Ig-like domain and MICA*008 depends on some ER-acquired factor, process, or modification (other than glycosylation trimming). Or, it might simply be that the formation of the US9-SEL1L-MICA*008 complex is inefficient. The answer to this question would also help explain how the Ig-like domain arrests MICA*008 maturation prior to the GPI-anchoring step, rather than indiscriminately retaining both unanchored and anchored MICA*008.

Another open question is why should the US9 SP be cleaved at all, if it is so important. One possible explanation comes from the observation that SP-retaining US9 forms were subject to faster turnover and were less abundant. Slow SP cleavage may therefore be a viral strategy to maximize MICA*008 degradation by balancing the greater efficacy but lower stability of the SP$^+$ form, with the lower efficacy but greater stability of the SP$^-$ form.

A different possibility is that SP removal may be required for other US9 functions. SP retention might affect US9 cellular localization and/or hamper its ability to target other proteins. US9 was recently shown[78] to impede the STING and MAVS pathways by direct association with STING and TBK1 in the ER, and by disrupting membrane integrity and MAVS signaling in the mitochondria. In that work, the cytoplasmic tail of US9, which we found to be redundant for MICA*008 targeting, was responsible for STING/MAVS targeting. It could be that SP cleavage is required for C-terminal mediated functions of US9, or for other as-yet-unknown functions.

In this context, previous work had shown that US9 localized to the mitochondria only when its SP was absent or dysfunctional[33]. It was not clearly indicated in the STING/MAVS study[78] whether the endogenous US9 SP was used, but it seems that at least the main construct used in that work, US9 attached to an N-terminal HA tag and a C-terminal EGFP, lacked the endogenous US9 SP. Furthermore, the N-terminal HA tag might have interfered with proper ER entry. This could explain why the HA-US9-EGFP construct showed partial mitochondrial localization, while US9 with its intact SP did not localize to the mitochondria in our hands or in previous studies[31,33,79]. Further studies are needed to

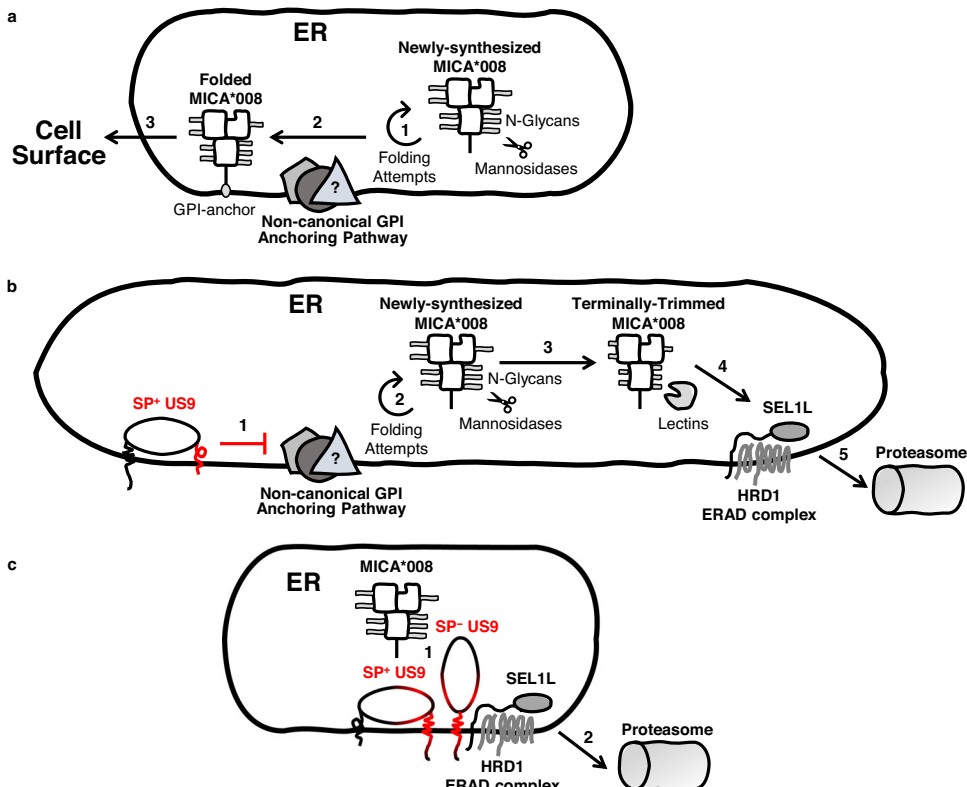

**Fig. 10 Unified model for US9 function. a** MICA*008's slow maturation process. Newly synthesized MICA*008 is recognized as misfolded by the ER quality control (ERQC) machinery, which retains it in the ER for repeated folding attempts and glycosylation trimming (1). Unknown factors (question mark) rescue MICA*008 from this cycle by slow delivery to the GPI-anchoring complex (2). GPI-anchored MICA*008 is recognized as properly folded, enabling its ER egress and expression on the cell surface (3). **b** US9 SP-dependent mode of function. The SP in the SP$^+$ US9 form (highlighted in red) interferes with the GPI-anchoring promoting factors (red line) resulting in MICA*008 maturation arrest (1). MICA*008 is retained in the ER by the ERQC machinery and undergoes repeated folding attempts and progressive glycosylation trimming (2). Eventually, MICA*008 become terminally trimmed and lectins deliver it to the SEL1L-HRD1 ER-associated degradation (ERAD) complex (3), which sends MICA*008 to proteasomal degradation (4). **c** US9 SP-independent mode of function. SP$^+$ US9 and SP$^-$ US9 bind MICA*008 through their luminal Ig-like domain, and SEL1L through their TM domain (domains are highlighted in red) (1). This interaction leads to SEL1L-HRD1 mediated retrotranslocation and proteasomal degradation of MICA*008 (2), with no dependence on glycosylation trimming.

understand US9 localization and the SP's role in it, especially during HCMV infection.

A major remaining mystery is the identity of the US9 SP cellular interaction partners. Studies are currently underway using the SP as a tool to uncover their identity and shed light on MICA*008's poorly understood maturation pathway. Beyond its value for basic cell biology, discovering MICA*008's maturation pathway may open possibilities for manipulating the expression of this common immune ligand, involved in autoimmunity, graft rejection, and cancer[20,21].

Finally, our findings have wider implications for our understanding of protein biology: namely, that US9's mode of function may not be unique, and similar cleavable effector SPs may play roles in various processes and organisms. US11 is a particularly attractive potential candidate that may have undescribed functions accomplished by its slowly cleaved SP, since no role has been found for it to date[32] (nor for US2's SP, which is entirely uncleaved[80]). Our results add to the mounting evidence that cleavable SPs play important roles beyond ER entry and cannot be regarded as interchangeable or inert.

## Methods

**Cells and antibodies**. RKO (CRL-2577) and HeLa (CCL-2) cell lines and MRC-5 primary lung fibroblasts (CCL-171), obtained from the ATCC, were used in this study. RKO and HeLa cells were kept in DMEM media supplemented with 10% FCS (Sigma-Aldrich, St. Louis, MO) and with 1% each of pen-strep, sodium

pyruvate, L-glutamine, and non-essential amino acids (Biological Industries, Beit Haemek, Israel). MRC-5 fibroblasts were kept in EMEM media with the same supplements and were used below passage 20. shRNA-transduced cells were grown in selection media containing 5 μg/ml puromycin (Merck Millipore, Billerica, MA). EYFP-transfected cells were grown in selection media containing 2 mg/ml G418 (Sigma-Aldrich). NK cells were isolated from peripheral blood lymphocytes using the EasySep human NK cell enrichment kit (StemCells Technologies, Vancouver, Canada). NK cells were co-cultured with irradiated allogeneic feeder cells, and activated using 500 U/ml rhIL-2 (PeproTech, Rehovot, Israel) and 20 μg/ml PHA (Roche, Basel, Switzerland). NK purity was >95% measured by flow cytometry staining for the CD56$^+$CD3$^-$ fraction. All primary cells were obtained in accordance with the institutional guidelines and permissions for using human tissues.

The anti-MICA (clone 159227, R&D Systems, Minneapolis, MN), anti-MICB (clone 236511, R&D Systems), anti-ULBP1 (clone 170818, R&D Systems), anti-ULBP2/5/6 (clone 165903, R&D Systems), anti-ULBP3 (clone 166514, R&D Systems), and anti-MHC class I (W6/32 hybridoma) primary antibodies were used for flow cytometry. The anti-PDI (rabbit polyclonal, Abcam, Cambridge, MA), anti-MICA (clone 159227, R&D Systems), and anti-His tag (clone AD1.1.10, R&D systems) primary antibodies were used for immunofluorescence.

The following secondary antibodies were used for flow cytometry and immunofluorescence: anti-mouse AlexaFluor 647 and anti-rabbit Cy3, all purchased from Jackson Laboratories (Bar Harbor, ME).

The following primary antibodies were used for immunoblotting: anti-His tag (clone AD1.1.10, R&D systems), anti-MICA (clone EPR6568, Abcam), anti-SEL1L (rabbit polyclonal to N-terminus, Sigma-Aldrich), EYFP cross-reactive anti-GFP (rabbit polyclonal, ab290, Abcam), anti-HRD1 (rabbit polyclonal, NB100-2526, Novus Biologicals, Centennial, CO), anti-ubiquitin (clone FK2, Merck Millipore), anti-GAPDH (clone 6C5, Santa Cruz, Dallas, TX), and anti-vinculin (clone EPR8185, Abcam).

The following secondary antibodies were used for immunoblotting: anti-mouse-HRP and anti-rabbit-HRP, light and heavy chain specific, or light chain specific, all purchased from Jackson laboratories.

**PCR mutagenesis, plasmid transfection, and lentiviral transduction**. The US8-HIS and US9-HIS constructs were previously described[28], and were used as templates for PCR mutagenesis. A table detailing the primers and templates used for each reaction is included in the Supplementary Information file. For EYFP constructs, PCR fragments were amplified and cloned into pIRES-neo FLAG/HA EYFP (Addgene #10825; kind gift of Dr. K. Le-Trilling, University Hospital Essen, Germany) using the EcoRV and NotI restriction sites. Plasmids were then directly transfected into cells using TransIT-LT1 Transfection reagent (Mirus). All other constructs were cloned into the pHAGE-DsRED(−)-eGFP(+) lentiviral vector using the NotI and BamHI restriction sites. All constructs were validated by DNA sequencing. Lentiviruses were generated in 293T cells using a transient three-plasmid transfection protocol[81]: plasmids containing the insert, Gag-pol, and PDMG were transfected and supernatants containing lentivirusal progeny were harvested 48 h later, centrifuged to eliminate cell debris and stored at −80 °C. Transduction was performed in the presence of Polybrene (6 μg/ml). Transduction efficiency was assessed by fluorescence levels, and only cell populations with >90% efficiency were used for experiments. Repeat transduction was performed if required to obtain the desired efficiency.

**MICA genotyping**. Genomic DNA was extracted from MRC-5 fibroblasts using the AccuPrep genomic DNA extraction kit (Bioneer, Daejeon, South Korea) according to the manufacturer's instructions. MICA Exons 2–5 were amplified using previously described primers (FW: CGTTCTTGTCCCTTTGCCCGTGTGC and REV: GATGCTGCCCCATTCCCTTCCCAA)[28,82], and the resulting PCR products were then ligated into pGEM T-easy plasmids (Promega, Madison, WI) and sent for sequencing. Sequences were compared to all known MICA alleles in the IMGT/HLA database (http://www.ebi.ac.uk/ipd/imgt/hla/) and verified by manually annotating exons and aligning them to the candidate alleles using global sequence alignment (emboss.bioinformatics.nl/). At least four separate sequencings were performed before a cell line was deemed to be homozygous.

**HCMV infection**. The US9 deletion mutant (ΔUS9) was previously generated using the BAC-cloned AD169varL genome pAD169[28]. Virus stocks were grown on human foreskin fibroblasts, titrated using a plaque assay, and stored at −80 °C. Infection was carried out at a multiplicity of infection (MOI) of 2.5, in confluent MRC-5 fibroblasts. HCMV infection was enhanced by centrifugation at 800 × g for 30 min. Mock-infected fibroblasts were treated with medium only, and were plated at a lower density so they would be sub-confluent at the time point in which the infected cells were harvested. Infection was verified by intracellular flow cytometry staining with anti-CMV antibody (clone 8B1.2, Merck Millipore) at 24 hpi, and at least 85% of the cells were infected.

**SEL1L and HRD1 knockdown**. Five SEL1L MISSION shRNA clones were purchased from Sigma-Aldrich for each targeted gene, expressed using lentiviral transduction as described above, and screened by immunoblot for knockdown efficiency in RKO MICA*008-HA cells. Two SEL1L shRNAs, designated as #1 (TRCN0000161385) and #2 (TRCN0000165954), and two HRD1 shRNAs, designated as #3 (TRCN0000364475) and #4 (TRCN0000364477), were chosen for subsequent experiments.

**Immunofluorescence**. Cells were grown on glass slides and fixed and permeabilized in cold (−20 °C) methanol. Cells were blocked overnight in CAS-block (Life Technologies, Carlsbad, CA), then incubated overnight with primary antibodies diluted 1:200 in CAS-block, then washed and incubated overnight in secondary antibodies diluted 1:500 in 5% BSA PBS. Cells were then washed, treated for 5 min with DAPI, and covered with coverslips. A confocal laser scanning microscope (Zeiss Axiovert 200M; Carl Zeiss MicroImaging, Thornwood, New York) was used to obtain images. Images were acquired and processed using the Olympus FluoView FV1000 software.

**Flow cytometry**. For flow cytometry, cells were plated at equal densities and incubated overnight. Resuspended cells were incubated on ice for 1 h with the primary antibody at a concentration of 0.2 μg per well. The cells were then incubated for 30 min on ice with the appropriate secondary antibody at a concentration of 0.75 μg per well. In all experiments using cells transduced with a GFP-expressing lentivirus or transfected with EYFP, the histograms are gated on the live GFP/EYFP+ population. Gating strategy is described in Fig. S2f. 10,000 live cells were acquired from each sample. Background stainings with a secondary antibody alone or with an isotype-matched control in HCMV-infected cells were performed for all cell types in the experiment. A single representative control staining (identified in the figure legends) is shown for each experiment, and all background stainings were similar to the one shown. Flow cytometry data were analyzed using FCS Express version 6 (De Novo Software, Pasadena, CA).

**CD107a-degranulation assay**. Analysis of CD107a on NK cell surface was previously described[51]. Briefly, primary IL-2-activated bulk NK cells were co-incubated with the target cells at a ratio of 1:1 at 37 °C for 2 h, in the presence of an APC-conjugated CD107a antibody and a PE-conjugated CD56 antibody (Biotest). CD107a levels on NK cells were then determined by flow cytometry. Three separate

NK donors were used for the experiments. Degranulation in the presence of EV targets was set as 100%, and the normalized reduction in degranulation compared to the EV control was calculated for each mutant.

**Immunoblot**. Cells were plated at an equal density, incubated overnight, and lysed in buffer containing 0.6% SDS and 10 mM Tris (pH 7.4) supplemented with 1 mM PMSF, 1:100 aprotinin (Sigma-Aldrich). In certain cases, lysates were digested with endoH or PNGaseF (New England Biolabs, Ipswich, MA), according to the manufacturer's instructions. Sodium dodecyl sulfate–polyacrylamide gel electrophoresis was performed using 10, 12.5, or 15% gels. Proteins were then blotted onto nitrocellulose membranes. Membranes were blocked for 1 h at room temperature (PBS, 5% skim milk, 0.4% Tween) and stained using the primary and secondary antibodies described above. Membranes were developed using the EZ-ECL kit (Biological Industries). Images were acquired and quantified using the Image Lab software (Bio-Rad, Hercules, CA), and blot contrast and brightness was adjusted as appropriate for better visualization. Unmodified full blots are included as a separate file.

**NP40 fractionation**. Cells were plated at an equal density, incubated overnight, and lysed in 1% NP40 lysis buffer (150 mM NaCl, 50 mM Tris, pH 7.4, 1 mM PMSF, 1:100 aprotinin). The lysates were centrifuged at 12,000 × g for 10 min and the supernatants were collected as the detergent-soluble fraction. Pellets were washed three times in PBS and then resuspended in protein sample buffer, boiled for 10 min at 99 °C, and collected as the detergent-insoluble fraction.

**Cycloheximide chase**. For cycloheximide chase, cells were incubated for 4–5 h in the presence of cycloheximide (50 μg/ml, Sigma-Aldrich) in combination with epoxomicin (Merck Millipore) or with DMSO mock treatment. Lysates were prepared for immunoblotting as described above at 0 h and at subsequent time points during the chase.

**Kifunensine treatment**. Cells were plated in a 6-well plate at a density of 400,000 cells per well. The following day, cells were treated with kifunensine (Cayman Chemical, Ann Arbor, MI) at a concentration of 200 μM or with equivalent concentrations of mock treatment (ultra-pure water). Cells were incubated for 24 h and then harvested for immunoblot analysis as described above.

**Immunoprecipitation**. RKO MICA*008-HA cells expressing US9, ΔIg, ΔSer, or ΔN-Ser were plated at equal densities to reach 90% confluence the following day. Cells were then incubated at 37 °C overnight with 50 nM epoxomicin (Merck Millipore). Cells were washed twice in ice-cold PBS and lysed for 30 min on ice in 1% NP40 lysis buffer (150 mM NaCl, 50 mM Tris, pH 7.4) supplemented with mammalian protease inhibitor cocktail (1:100, Sigma-Aldrich). Following lysis, nuclei and debris were cleared by centrifugation (10,000 × g, 4 °C, 20 min). Supernatants were pre-cleared for 30 min at 4 °C using an isotype-matched control (mouse IgG1, Biolegend, San Diego, CA) and protein G-plus beads (Santa-Cruz). Beads were then removed by centrifugation (1000 × g, 4 °C, 5 min). The pre-cleared supernatants were then mixed with an anti-His tag antibody (AD1.1.10, R&D Systems), and incubated for 1 h at 4 °C on a rotating platform. After 1 h, protein G-plus beads were added for an overnight incubation at 4 °C on a rotating platform. The following day, four washes were performed in ice-cold high-salt 0.2% NP40 wash buffer (500 mM NaCl, 50 mM Tris, pH 7.4) followed by two washes in ice-cold PBS and eluted by boiling for 3 min in twofold concentrated protein sample buffer. The eluates were run in gel electrophoresis on 15% gels which were then stained with Imperial protein stain (Thermo Fisher Scientific, Waltham, MA). Immunoprecipitated bands were excised based on size. For US9, the two isoforms were excised separately. Excised bands were shipped to the Smoler Proteomics Center, Department of Biology, Technion Institute of Technology, analyzed by LC-MS/MS on Q-Exactive (Thermo Fisher Scientific) and identified by Discoverer software against the US9 sequence. All the identified peptides were filtered with high confidence, top rank, mass accuracy, and minimum of two peptides. High confidence peptides have passed the 1% FDR threshold. Semi quantitation was done by calculating the peak area of each peptide. The area of the protein is the average of the three most intense peptides from each protein.

HA-tag immunoprecipitation was performed using anti-HA tag (clone 12B16, Abcam) and protein G-plus beads. RKO MICA*008-HA-US9 cells expressing either shScrambled control or shHRD1#4 were plated overnight as described above and incubated for 5 h with 4 μM of the proteasomal inhibitor EPX. Washing, lysis, preclearance, high-salt washing, and elution were performed as described above.

**Co-immunoprecipitation**. Cells were plated at equal densities to reach 90% confluence the following day. Cells were then incubated for 4 h at 37 °C with 2 μM epoxomicin, washed twice in ice-cold PBS, and lysed for 30 min on ice in 1% digitonin lysis buffer (150 mM NaCl, 50 mM Tris, pH 7.4) supplemented with mammalian protease inhibitor cocktail (1:100, Sigma-Aldrich). Following lysis, nuclei and debris were cleared by centrifugation (10,000 × g, 4 °C, 20 min). Input controls were taken from the supernatants after this stage.

For anti-His tag co-immunoprecipitation, supernatants were pre-cleared for 30 min at 4 °C using an isotype-matched control (mouse IgG1, Biolegend) and

protein G-plus beads (Santa-Cruz). Beads were then removed by centrifugation ($1000 \times g$, 4 °C, 5 min). The pre-cleared supernatants were then mixed with an anti-His tag antibody (AD1.1.10, R&D Systems), and incubated for 1 h at 4 °C on a rotating platform. After 1 h, protein G-plus beads were added for an overnight incubation at 4 °C on a rotating platform. The following day, four washes were performed in ice-cold 0.2% digitonin wash buffer (150 mM NaCl, 50 mM Tris, pH 7.4) followed by two washes in ice-cold PBS. For immunoblotting, beads were eluted by boiling in twofold concentrated protein sample buffer for 3 min.

For proteomics, the same anti-His immunoprecipitation protocol was used, except that two controls were used: empty-vector-transduced cells immunoprecipitated with an anti-His antibody, and US9-HIS-transduced cells immunoprecipitated with an isotype control. Uneluted beads were stored at −80 °C and shipped to the Smoler Proteomics Center, Department of Biology, Technion Institute of Technology, where LC-MS/MS was performed on Orbitrap XL (Thermo Fisher Scientific), and results were analyzed by the MaxQuant software. All the identified peptides were filtered with 1% FDR threshold and minimum of two peptides. Known contaminants were also excluded.

For anti-HA tag co-immunoprecipitation, covalently attached anti-HA agarose beads (clone HA-7, Sigma-Aldrich) were used according to the manufacturer's instructions. Elution was performed by boiling in twofold concentrated protein sample buffer for 3 min.

**Metabolic labeling**. Cells grown in 6-well plates were washed with PBS and metabolically labeled (Easytag Express [35S]-Met/Cys protein labeling, Perkin Elmer, Waltham, MA) with 100 Ci/ml for 20 min. Cells were lysed in digitonin lysis buffer (140 mM NaCl, 20 mM Tris [pH 7.6], 5 mM MgCl$_2$, and 1% digitonin (Merck Millipore)) and cleared from membrane debris at 13,000 rpm for 30 min at 4 °C. Lysates were incubated with anti-His tag antibody for 2 h at 4 °C in an overhead tumbler before immune complexes were retrieved by protein G-sepharose (GE Healthcare, Chicago, IL). Sepharose pellets were washed four times with increasing NaCl concentrations (0.15–0.5 M in lysis buffer containing 0.2% detergent). EndoH (New England Biolabs) treatment was performed as recommended by the manufacturer. Prior to loading onto a SDS-PAGE immune complexes were dissociated at 95 °C for 5 min in a DTT (40 mM) containing sample buffer. Fixed and dried gels were exposed overnight to a phosphor screen, scanned by Typhoon FLA 7000 (GE Healthcare). For better visualization of the results, contrast and brightness were adjusted. A long exposure to X-ray film was used for autoradiography. Images were quantified using the ImageJ software[83].

**Statistical methods**. A one-way ANOVA was used to compare the effect of the various US9 mutants and chimeras on MICA surface expression (measured as normalized median fluorescent intensity). All mutants sharing the same control groups (e.g., all single US9 mutants, all double US9 mutants, all EYFP chimeras, etc.) were grouped into the same analysis to correct for the relevant multiple comparisons. EV controls were not included in the analyses. Where the ANOVA was statistically significant, a post hoc Dunnett's test was conducted to determine which mutants differed significantly from the control group. A two-way ANOVA was used to compare the effects of SEL1L knockdown and the various US9 constructs, including EV controls, on MICA surface expression, and where the ANOVA was statistically significant, a post hoc Tukey test was conducted to determine which mutants differed significantly from each other. ANOVAs and post hoc tests were considered statistically significant when $p < 0.05$. Full statistical details including $p$ values, $F$ values, degrees of freedom, and effect sizes (Cohen's $d$) are presented in the relevant figure legends and/or in the Source data file.

For specific comparison of the effect of US9 on MICA surface and total protein levels in shSCR, shSEL1L, or shHRD1 KD cells, a two-tailed Student's $t$-test for unequal variance samples was used, with Sidak's correction for multiple comparisons. The t-test was considered significant when $p < [1 - (1 - 0.05)^{\frac{1}{m}}]$, where $m$ equals the number of null hypotheses tested. Results from the two shRNA constructs targeting the same gene (SEL1L or HRD1) were pooled together in this analysis.

Where multiple comparisons were not needed, a two-tailed Student's $t$-test for unequal variance samples was used without corrections and was considered statistically significant when $p < 0.05$.

The number of independent experiments analyzed (biological replicates) is indicated in the relevant figure legends. Biological replicates were performed with fresh cells from at least two separate transfections/transductions. The data analysis for this paper was generated using the Real Statistics Resource Pack software (Release 5.4). Copyright (2013–2018) Charles Zaiontz. www.real-statistics.com, and GraphPad Prism version 8 for Windows, GraphPad Software, La Jolla, California, USA, www.graphpad.com.

**Reporting summary**. Further information on research design is available in the Nature Research Reporting Summary linked to this article.

## Data availability
The data underlying Figs. 1c, 3b, 3d, 4d, 5c, 7c, 7e, 8a, 9c, 9e, and Supplementary Figs. 4d, 7d, 8d and 8f, including full statistical analyses, are provided as a Source data file. Remaining data are available in the article, Supplementary data files or available from the authors upon request.

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

## Acknowledgements

This study was supported by the ISF Israel-China grant. Further support came from the GIF foundation, the ICRF professorship grant, the Israeli Science Foundation (Moked), a Ministry of Science Personal Medicine grant, and the DKFZ-MOST grant. E.S. was

supported during her work by the Adams Fellowship Program of the Israel Academy of Sciences and Humanities and by the Foulkes Foundation.

## Author contributions

E.S. and O.M. conceived and planned the experiments, in consultation with B.T. and A.H. E.S., L.D., S.K. and A.H. performed the experiments and analyzed the data. T.S.M. performed statistical analyses. O.M. supervised the project. All authors discussed the results and contributed to the final manuscript.

## Competing interests

The authors declare no competing interests.
