## [Peer Review File · Nature Communications]

Reviewers' Comments:

Reviewer #1:

Remarks to the Author:

The manuscript by Seidel et al. delineated the molecular mechanism by which the HCMV glycoprotein US9 downregulates the expression of MICA*008, a ligand that is expressed on the host cell surface and important for NK cell-mediated antiviral immune response. Specifically, the authors demonstrate a surprising role played by N-terminal signal peptide (SP) of US9 in the turnover of MICA*008. Interestingly, these authors show SP is only slowly cleaved and SP+ US9 isoform, not the SP- US9 isoform, downregulate surface MICA*008. Mechanistically, these authors show Sel1L-Hrd1 ERAD plays an important role in the degradation of the ER-resident MICA*008.

Overall, this study presents some important, interesting and novel finding and is, for the most part, solid. The experiment designs are rigorous and sophisticated. The authors comprehensively analyzed the functional domains of US9 protein and delineated the nature of SP cleavage. They convincingly demonstrated that SP is both required and sufficient for down-regulation of cell surface MICA*008. The authors provided sufficient experimental details and stats analyses.

I have a few comments, centered on SEL1L-HRD1 ERAD, that should be considered during the revision.

1. Most importantly, can the authors confirm the size of SEL1L band? It's a 90kDa protein, and in our hand, it runs right below 100kDa. However, in this paper, it runs around 75kda. Can the authors use the Ab from Abcam to confirm the results? A side-by-side comparison of the Abcam and Sigma Ab (used in this study) would be sufficient. I remembered the Sigma Ab did not work in our hand. We also recently generated SEL1L Ab for IP – which we are happy to share.
2. Another important point is that there is no data showing that HRD1 is involved. SEL1L has been reported to have HRD1 independent function. Hence, it will be necessary and critical to show that HRD1 mediate ubiquitination of MICA*008 in a ligase activity dependent manner. This data is required to support the notion that MICA*008 is a bona fide substrate of SEL1L-HRD1 ERAD.
3. Additionally, where is the fate of accumulated MICA*008 in SEL1L KO cells? The authors should consider doing IF to visualize MICA distribution or perform biochemical assays (non-denaturing gels or sucrose gradient fractionation) to assay the formation of HMW – as described in several recent SEL1L papers.
4. Many of recent papers on the physiological role of SEL1L-HRD1 ERAD and the fate of endogenous substrates were left out.

Reviewer: Ling Qi

Reviewer #2:

Remarks to the Author:

Natural killer (NK) cells are crucial for controlling Human Cytomegalovirus (HCMV) infection, and therefore, HCMV must evade the NK cell response in order to establish infection. Many HCMV proteins are involved in efficient evasion of MHC class I and NK cell responses, and many novel insights into cell biology could be obtained by studying these viral antagonists in the past.

The manuscript by Seidel et al. describes two distinct mechanisms how the US9 protein of HCMV evades natural killer (NK) cell responses mediated by the activating NK cell receptor NKG2D and its activating ligand MHC class I polypeptide related sequence A (MICA). This group has previously shown that US9 targets MICA, more precisely MICA*008, which is the most prevalent allele of this polymorphic family (Seidel et al 2015, Cell Reports). MICA is an activating ligand expressed on the cell surface of the infected cells and is recognized/bound by the activating NK cell receptor NKG2D. To

reach the cell surface and allow recognition by NKG2D, MICA needs to be synthesized in the ER and traffic to the plasma membrane. MICA*008 has distinct biological properties: it is first synthesized as a truncated soluble protein and is subsequently equipped with a GPI-anchor in the ER, which then allows ER egress. Intriguingly, MICA*008 lacks a canonical GPI-anchoring signal, and it is unknown which proteins of the GPI-anchoring machinery or contributing unknown factors are involved in this process. Altogether, this non-standard maturation process is rather slow compared to other MICA alleles, which are transmembrane proteins and leave the ER much faster.

While Seidel et al. already identified US9 as an antagonist of MICA*008 in 2015, they did not show the exact mechanism how immune evasion was achieved. They could already show that US9 targets MICA*008 for proteasomal degradation, dependent on MICA*008's non-standard maturation pathway, and this occurred with slow kinetics prior to the GPI anchoring step. Another prerequisite for US9 function was that MICA*008 undergoes non-canonical GPI anchoring. However, the signal peptide (SP) of US9 had not been implicated into evasion from MICA*008/NKG2D.

Now, they describe two mechanisms how US9 inhibits MICA*008 trafficking, and they provide an extensive amount of data to support their findings.

First, they describe the phenomenon that cleavage of the signal peptide (SP) of US9 occurs very slowly. Then, they show that the SP of US9 is the major domain of US9 that is responsible for targeting MICA*008 for degradation (which is only possible because it is cleaved with such slow kinetics). This is indeed a novel finding – I am not aware of any study that linked the signal peptide of a protein to any kind of immune evasive function (which is understandable since they are usually rapidly cleaved).

Second, the authors show that US9 employs another mechanism to target MICA*008, and this is independent of the SP: they identified the cellular SEL1L protein as an interacting partner of US9 by qAP-MS and could demonstrate that US9, by directly interacting with MICA*008 and SEL1L, delivers MICA*008 for degradation. SEL1L is a component of the ER-associated degradation (ERAD) complex. Third, they show that US9 arrests MICA*008 maturation via its SP and indirectly induces degradation mediated by the ER quality control compartment (ERQC) via the SEL1L-HRD1 ERAD complex. Overall, this study is well done, their findings are very interesting and the role of the US9 SP for immune evasion is novel. These findings are certainly interesting for the field of virology, immunology, and cell biology.

The manuscript is written very well, and I appreciate very much that the authors clearly point out the missing pieces of the puzzle in their discussion. Also, I appreciate that results are not overinterpreted. The manuscript is an enjoyable read. I think that the authors gave enough details to allow reproducibility of their work.

But here comes the famous "However". What I miss from the manuscript are novel insights about cell biological processes – it would be fantastic to understand how the US9 SP influences the GPI-anchoring step of MICA*008. This GPI-anchoring step is poorly understood and US9 would be the perfect tool to obtain novel insights into this process. As the authors already suggest by themselves in their discussion, qAP-MS experiments with the appropriate constructs (US9 SP-/+, US8 with US9's SP, etc.) have the potential to revolutionize the field, since the factors that confer GPI-anchoring of MICA*008 are not known. I know that I am asking for many more experiments here, and I am aware of the high amount of data that has been produced already, but this would really add a lot to the field of cell biology and would address a larger audience.

Minor comments:

1. Figure 1: It would be nice to quantify panel B (SP+, SP-) to underline the statement in line 142-143 (SP+ decreased more rapidly than the SP- form accumulated).
2. Figure 1: Panel C at first sight looks like a scheme of the US9 protein showing different domains. It would be nice to add another scheme to it that shows how the experiment was actually done to make clear that this scheme shows peptide sequence coverage. Could the authors also please elaborate a bit on how this experiment was done? Were bands (SP+, SP-) excised from an

electrophoresis gel?

Wording line 150: not sure this should be called sequencing – you detect peptides, but you do not really sequence the protein.

3. Lines 203-204: this sentence needs to be moved to figure 1. Can the authors please comment if RKO cells express and endogenous MICA*008? I assume not, but it would be nice to read this explicitly somewhere (I may have overlooked it though).

4. I have only minor comments on the written presentation of the manuscript:

- In my opinion, immunoblot/immunoblotting is a better term than western blot.
- FACS stands for cell sorting – were cells really sorted, or “just” analysed by flow cytometry?

Point by Point

Reviewer #1

Comment: The manuscript by Seidel et al. delineated the molecular mechanism by which the HCMV glycoprotein US9 downregulates the expression of MICA*008, a ligand that is expressed on the host cell surface and important for NK cell-mediated antiviral immune response. Specifically, the authors demonstrate a surprising role played by N-terminal signal peptide (SP) of US9 in the turnover of MICA*008. Interestingly, these authors show SP is only slowly cleaved and SP+ US9 isoform, not the SP- US9 isoform, downregulate surface MICA*008. Mechanistically, these authors show Sel1L-Hrd1 ERAD plays an important role in the degradation of the ER-resident MICA*008.

Overall, this study presents some important, interesting and novel finding and is, for the most part, solid. The experiment designs are rigorous and sophisticated. The authors comprehensively analyzed the functional domains of US9 protein and delineated the nature of SP cleavage. They convincingly demonstrated that SP is both required and sufficient for down-regulation of cell surface MICA*008. The authors provided sufficient experimental details and stats analyses.

Response: We thank the reviewer for the comments and the positive overall assessment of our work. We replied to all criticisms and changed the manuscript accordingly (changes are marked in red).

Comment: I have a few comments, centered on SEL1L-HRD1 ERAD, that should be considered during the revision.

Response: We have performed the additional experiments regarding SEL1L-HRD1 ERAD requested by the reviewer, and we hope we now address any remaining concerns the reviewer may have.

Comment: 1. Most importantly, can the authors confirm the size of SEL1L band? It's a 90kDa protein, and in our hand, it runs right below 100kDa. However, in this paper, it runs around 75kda. Can the authors use the Ab from Abcam to confirm the results? A side-by-side comparison of the Abcam and Sigma Ab (used in this study) would be sufficient. I remembered the Sigma Ab did not work in our hand. We also recently generated SEL1L Ab for IP – which we are happy to share.

Response: The reviewer is correct in pointing out that SEL1L is 90 kDa protein. Accordingly, we observed a running size of ~90-100 kDa in all the figures in our manuscript. However, in figure 6B where there was an error in labelling so that the marker of 100 kDa was accidentally labelled as 75 kDa, causing the running size to appear smaller than it actually is. We believe this error is the reason for the reviewer's misgivings. This error has now been corrected, and we thank the reviewer for bringing it to our attention!

To demonstrate the running size of SEL1L in 10% and 15% polyacrylamide gels, we attach two uncropped blots of SEL1L. All uncropped blots shown in this manuscript are included as a separate supplementary file.

Regarding the validity of the sigma Aldrich antibody used in this study, it has been used in multiple high impact publications, including in Nature Communications. Here are a few examples:

- van de Weijer, M., Bassik, M., Luteijn, R. *et al.* A high-coverage shRNA screen identifies TMEM129 as an E3 ligase involved in ER-associated protein degradation. *Nat Commun* **5**, 3832 (2014). <https://doi.org/10.1038/ncomms4832>
- Metformin Promotes Antitumor Immunity via Endoplasmic-Reticulum-Associated Degradation of PD-L1. *Mol Cell* 2018 Aug 16;71(4):606-620.e7. doi: 10.1016/j.molcel.2018.07.030.
- Ye, Y., Baek, S., Ye, Y. *et al.* Proteomic characterization of endogenous substrates of mammalian ubiquitin ligase Hrd1. *Cell Biosci* **8**, 46 (2018). <https://doi.org/10.1186/s13578-018-0245-z>
- Human Cytomegalovirus Tropism Modulator UL148 Interacts with SEL1L, a Cellular Factor That Governs Endoplasmic Reticulum-Associated Degradation of the Viral Envelope Glycoprotein gO. *JVI* Aug 2018, 92 (18) e00688-18; DOI: 10.1128/JVI.00688-18
- Down-modulation of SEL1L, an unfolded protein response and endoplasmic reticulum-associated degradation protein, sensitizes glioma stem cells to the cytotoxic effect of valproic acid Cattaneo M, *et al.* *JBC* 289(5), 2826-2838, (2014) DOI: 10.1074/jbc.M113.527754

Therefore, and since the aforementioned size discrepancy was due to a simple error of labelling, we feel confident in the validity of the antibody we used. Furthermore, the fact that two different and externally validated shRNA constructs resulted in the disappearance of the correctly-sized band we observed (Fig. S7A), also supports the specificity of this antibody.

However, if the reviewer is still interested in a side-by-side comparison with another anti-SEL1L antibody, we will be happy to perform it. We also want to express our gratitude for the reviewer's kind offer to share reagents with us.

Comment: 2. Another important point is that there is no data showing that HRD1 is involved. SEL1L has been reported to have HRD1 independent function. Hence, it will be necessary and critical to show that HRD1 mediate ubiquitination of MICA*008 in a ligase activity dependent manner. This data is required to support the notion that MICA*008 is a bona fide substrate of SEL1L-HRD1 ERAD.

Response: To address the question of HRD1's involvement in US9-mediated MICA*008 degradation, we generated HRD1 knockdown cells (Fig. S8A-B), and assayed the effect of HRD1 depletion on MICA*008 surface expression (Fig. S8C-D) and total protein levels (Fig. S8E-F), in the absence or presence of US9. We found that like SEL1L knockdown, HRD1 knockdown also results in a moderate (~30-50%) increase in MICA*008 levels, suggesting MICA*008 is an endogenous

substrate of HRD1-mediated ERAD. In the presence of US9, HRD1 depletion did not restore surface MICA*008 expression, but did rescue total MICA*008 protein levels and caused the accumulation of a glycosylation-trimmed form of MICA*008. This shows that in HRD1's absence, US9-mediated degradation of MICA*008 was impaired, and instead MICA*008 accumulated intracellularly. Crucially, this is identical to the phenotype we observed with SEL1L knockdown cells, strongly supporting the hypothesis that both SEL1L and HRD1 are required for MICA*008 degradation. Finally, to show that HRD1 mediates MICA*008 ubiquitination, we immunoprecipitated MICA*008 from US9-expressing cells transduced a control shRNA or with shHRD1 (Fig. S8 G-H). We found that the relative ratio of ubiquitinated MICA*008 to total MICA*008 was reduced by ~70% in HRD1 depleted cells, showing that MICA*008 is a bona-fide substrate of the HRD1-SEL1L complex. These results are described in detail in lines 547-575 of the revised manuscript and shown in figure S8.

Comment: 3. Additionally, where is the fate of accumulated MICA*008 in SEL1L KO cells? The authors should consider doing IF to visualize MICA distribution or perform biochemical assays (non-denaturing gels or sucrose gradient fractionation) to assay the formation of HMW – as described in several recent SEL1L papers.

Response: We performed immunofluorescence visualizing MICA distribution in SEL1L knockdown cells (Fig. S7E, described in lines 483-486 and 530-533). We observed a uniform distribution of MICA*008, without any puncta suggestive of HMW complexes. To directly address this point, we conducted an NP40 solubility assay (Fig. S7G, described in lines 533-545), and observed that ER-resident MICA largely remained in the NP40-soluble fraction even in SEL1L-depleted cells, suggesting SEL1L is not required for the prevention of MICA*008 aggregation.

Comment: 4. Many of recent papers on the physiological role of SEL1L-HRD1 ERAD and the fate of endogenous substrates were left out.

Response: We thank the reviewer for pointing out this omission. We initially focused only on SEL1L's historical role in the context of HCMV infection, and have now expanded this section to better encompass the current body of knowledge regarding SEL1L and its many functions in health and disease (lines 448-451, 523-527).

Reviewer: Ling Qi

Reviewer #2

Comment: Natural killer (NK) cells are crucial for controlling Human Cytomegalovirus (HCMV) infection, and therefore, HCMV must evade the NK cell response in order to establish infection. Many HCMV proteins are involved in efficient evasion of MHC class I and NK cell responses, and many novel insights into cell biology could be obtained by studying these viral antagonists in the past.

The manuscript by Seidel et al. describes two distinct mechanisms how the US9 protein of HCMV evades natural killer (NK) cell responses mediated by the activating NK cell receptor NKG2D and its activating ligand MHC class I polypeptide related sequence A (MICA). This group has previously shown that US9 targets MICA, more precisely MICA*008, which is the most prevalent allele of this

polymorphic family (Seidel et al 2015, Cell Reports). MICA is an activating ligand expressed on the cell surface of the infected cells and is recognized/bound by the activating NK cell receptor NKG2D. To reach the cell surface and allow recognition by NKG2D, MICA needs to be synthesized in the ER and traffic to the plasma membrane. MICA*008 has distinct biological properties: it is first synthesized as a truncated soluble protein and is subsequently equipped with a GPI-anchor in the ER, which then allows ER egress. Intriguingly, MICA*008 lacks a canonical GPI-anchoring signal, and it is unknown which proteins of the GPI-anchoring machinery or contributing unknown factors are involved in this process. Altogether, this non-standard maturation process is rather slow compared to other MICA alleles, which are transmembrane proteins and leave the ER much faster.

While Seidel et al. already identified US9 as an antagonist of MICA*008 in 2015, they did not show the exact mechanism how immune evasion was achieved. They could already show that US9 targets MICA*008 for proteasomal degradation, dependent on MICA*008's non-standard maturation pathway, and this occurred with slow kinetics prior to the GPI anchoring step. Another prerequisite for US9 function was that MICA*008 undergoes non-canonical GPI anchoring. However, the signal peptide (SP) of US9 had not been implicated into evasion from MICA*008/NKG2D.

Now, they describe two mechanisms how US9 inhibits MICA*008 trafficking, and they provide an extensive amount of data to support their findings.

First, they describe the phenomenon that cleavage of the signal peptide (SP) of US9 occurs very slowly. Then, they show that the SP of US9 is the major domain of US9 that is responsible for targeting MICA*008 for degradation (which is only possible because it is cleaved with such slow kinetics). This is indeed a novel finding – I am not aware of any study that linked the signal peptide of a protein to any kind of immune evasive function (which is understandable since they are usually rapidly cleaved).

Second, the authors show that US9 employs another mechanism to target MICA*008, and this is independent of the SP: they identified the cellular SEL1L protein as an interacting partner of US9 by qAP-MS and could demonstrate that US9, by directly interacting with MICA*008 and SEL1L, delivers MICA*008 for degradation. SEL1L is a component of the ER-associated degradation (ERAD) complex.

Third, they show that US9 arrests MICA*008 maturation via its SP and indirectly induces degradation mediated by the ER quality control compartment (ERQC) via the SEL1L-HRD1 ERAD complex. Overall, this study is well done, their findings are very interesting and the role of the US9 SP for immune evasion is novel. These findings are certainly interesting for the field of virology, immunology, and cell biology.

The manuscript is written very well, and I appreciate very much that the authors clearly point out the missing pieces of the puzzle in their discussion. Also, I appreciate that results are not overinterpreted. The manuscript is an enjoyable read. I think that the authors gave enough details to allow reproducibility of their work.

Response: We thank the reviewer for his kind comments regarding the manuscript's quality and level of interest. We replied to all criticisms and changed the manuscript accordingly (changes are marked in red).

Comment: But here comes the famous “However”. What I miss from the manuscript are novel insights about cell biological processes – it would be fantastic to understand how the US9 SP influences the GPI-anchoring step of MICA*008. This GPI-anchoring step is poorly understood and US9 would be the perfect tool to obtain novel insights into this process. As the authors already

suggest by themselves in their discussion, qAP-MS experiments with the appropriate constructs (US9 SP-/+, US8 with US9's SP, etc.) have the potential to revolutionize the field, since the factors that confer GPI-anchoring of MICA*008 are not known. I know that I am asking for many more experiments here, and I am aware of the high amount of data that has been produced already, but this would really add a lot to the field of cell biology and would address a larger audience.

Response: We entirely agree with the reviewer that understanding how the US9 SP influences MICA*008 GPI anchoring is of great interest to the field of cell biology. Indeed, we are pursuing precisely the line of inquiry suggested by the reviewer.

However, we feel that such a discovery is far beyond the scope of the current manuscript. Characterizing the unknown component(s) of the GPI-anchoring pathway would require far more than merely identifying them using mass spectrometry. Once identified, we are interested in analyzing their interaction with the US9 SP and with the GPI anchoring machinery; finding any additional proteins that depend on these components to confer GPI anchoring; and addressing the functional significance of these components both under physiological conditions and during HCMV infection.

We already have some promising preliminary results, and we sincerely hope we will be able to answer the important questions the reviewer raised here in the not too distant future.

Minor comments:

Comment: 1. Figure 1: It would be nice to quantify panel B (SP+, SP-) to underline the statement in line 142-143 (SP+ decreased more rapidly than the SP- form accumulated).

Response: We have added the quantification of the pulse chase data in Figure 1 panel C (lines 129-136), which supports our stated conclusion.

Comment: 2. Figure 1: Panel C at first sight looks like a scheme of the US9 protein showing different domains. It would be nice to add another scheme to it that shows how the experiment was actually done to make clear that this scheme shows peptide sequence coverage. Could the authors also please elaborate a bit on how this experiment was done? Were bands (SP+, SP-) excised from an electrophoresis gel?

Response: We annotated the figure (which is now Figure 1 panel D) to separately show US9 domains alongside the peptide sequence coverage, and hope that it is now clearer. The experiment was indeed performed by excision of the relevant bands from a polyacrylamide gel. While the procedure is described in detail in the Methods section, we also added further detail to the main text (lines 141-143)

Comment: Wording line 150: not sure this should be called sequencing – you detect peptides, but you do not really sequence the protein.

Response: The reviewer is correct in pointing this out. We now write that we "detected peptides derived from each band using mass spectrometry" (lines 142-143).

Comment: 3. Lines 203-204: this sentence needs to be moved to figure 1. Can the authors please comment if RKO cells express and endogenous MICA*008? I assume not, but it would be nice to read this explicitly somewhere (I may have overlooked it though).

Response: We added the following sentence to the section describing the generation of the cells used for figure 1: "We used previously described RKO cells which endogenously express very low levels of the full-length allele MICA*007:01, and overexpress MICA*008 fused to an N-terminal HA tag" (lines 114-116). We hope this point is now clearer.

Comment: 4. I have only minor comments on the written presentation of the manuscript:

- In my opinion, immunoblot/immunoblotting is a better term than western blot.
- FACS stands for cell sorting – were cells really sorted, or “just” analysed by flow cytometry?

Response: We edited the manuscript and now uniformly use the reviewer's suggested terminology of 'immunoblot' and 'flow cytometry' throughout, for greater accuracy and internal consistency.

Reviewers' Comments:

Reviewer #1:

Remarks to the Author:

the authors have addressed all of my concerns.